# Genome-wide CRISPR screenings identified *SMCHD1* as a host-restricting factor for AAV transduction

Chenlu Wang[1☯], Yu Liu[2☯], Jingfei Xiong[1], Kun Xie[1], Tianshu Wang[1], Yu Hu[1], Huancheng Fu[1], Baiquan Zhang[1], Xiaochao Huang[1], Hui Bao[1], Haoyang Cai[1], Biao Dong[2,3]*, Zhonghan Li[1]*

1 Center of Growth Metabolism and Aging, State Key Laboratory of Oral Disease, West China Hospital of Stomatology, Key Laboratory of Bio-Resource and Eco-Environment of Ministry of Education, Animal Disease Prevention and Food Safety Key Laboratory of Sichuan Province, College of Life Sciences, Sichuan University, Chengdu, China, 2 National Clinical Research Center for Geriatrics and State Key Laboratory of Biotherapy, West China Hospital, Sichuan University, Chengdu, China, 3 Sichuan Real and Best Biotech Co., Ltd., Chengdu, China

☯ These authors contributed equally to this work.
* Biaodong@scu.edu.cn (BD); Zhonghan.Li@outlook.com (ZL)

**Data Availability Statement:** All relevant data are within the manuscript and its Supporting Information files.

## Abstract

AAV-mediated gene therapy typically requires a high dose of viral transduction, risking acute immune responses and patient safety, part of which is due to limited understanding of the host-viral interactions, especially post-transduction viral genome processing. Here, through a genome-wide CRISPR screen, we identified *SMCHD1* (Structural Maintenance of Chromosomes Hinge Domain 1), an epigenetic modifier, as a critical broad-spectrum restricting host factor for post-entry AAV transgene expression. *SMCHD1* knock-down by RNAi and CRISPRi or knock-out by CRISPR all resulted in significantly enhanced transgene expression across multiple viral serotypes, as well as for both single-strand and self-complementary AAV genome types. Mechanistically, upon viral transduction, SMCHD1 effectively repressed AAV transcription by the formation of an LRIF1-HP1-containing protein complex and directly binding with the AAV genome to maintain a heterochromatin-like state. *SMCHD1*-KO or *LRIF1*-KD could disrupt such a complex and thus result in AAV transcriptional activation. Together, our results highlight the host factor-induced chromatin remodeling as a critical inhibitory mechanism for AAV transduction and may shed light on further improvement in AAV-based gene therapy.

## Author summary

Adeno-associated virus (AAV) is one of the most commonly used carrier for gene therapy. However, current approved therapies typically use extremely high doses in order to achieve sufficient beneficial efficiency. This is partially due to a lack of understanding about host reponsese that restrict AAV-mediated gene delivery and expression. In this study, we carried out genome-wide CRISPR screens to identify potential host restricting

**Funding:** This work was supported by the National Key Research and Development Program of China (2022YFA1104401 for H.C., 2021YFA1100601 for Z.L.), National Natural Science Foundation of China (32071455 for Z.L.), Key Research and Development Program of Sichuan Province (2021ZDZX0010 for Z.L.), SCU grant (020SCUNL109 for Z.L.), Research and Develop Program, West China Hospital of Stomatology Sichuan University (RD-03-202106 for Z.L.) and the Fundamental Research Funds for the Central Universities (SCU2019D013 for Z.L.). The funders had no role in study design, data collection and analysis, decision to publish, or preparation of the manuscript.

**Competing interests:** Prof. BD is the founder of Sichuan Real and Best Biotech Co., Ltd., Chengdu, China.

factors for AAV transduction in human cells and identified SMCHD1 as one of the key epigenetic modifers that formed a complex with LRIF1, bound with HP1-associated AAV genome to form a heterochromatin-like states, and suppressed its transcription. Disruption of the SMCHD1-LRIF1 complex would increase the genome accessibility of AAVs and viral transduction, and significantly increase AAV-mediated transgene expression. Therefore, our results indicated that the host factor-induced chromatin remodeling might be a critical inhibitory mechanism for AAV transduction and may help to further improve AAV-based gene therapy.

## Introduction

Adeno-associated virus (AAV) is widely used for gene therapy because of its simple genome structure, broad tissue tropism, non-pathogenic nature, and ability to achieve long-term transgene expression in low-proliferating cells [1]. In recent clinical trials, AAV-based therapeutics have been proven effective for several human genetic disorders, including lipoprotein lipase deficiency [2], hemophilia [3], and Duchene muscular dystrophy (DMD) [4] etc. However, extremely high vector doses are typically used to achieve robust transgene expression and treat human patients. For instance, ROCTAVIAN (AAV5) was recently approved by the FDA to treat adult hemophilia A with a dose of $6x10^{13}$ vg/kg [5], while ZOLGENSMA (AAV9) for spinal muscular atrophy was approved at $1.1x10^{14}$ vg/kg [6]. Given that the adult human body only contains $3x10^{13}$ cells in total [7], direct intravenous (I.V.) infusion of such a high load would, not uncommonly, cause severe adverse effects in patients, including hepatoxicity [8], thrombotic microangiopathy [9], and even patient deaths [10,11]. Therefore, there is an urgent need to develop AAV-based therapies at lower required dosages while maintaining delivery efficiency. However, such enthusiasm is often dampened by the lack of a thorough understanding of AAV transduction in human cells.

AAV transduction can be divided into several steps, including membrane binding and internalization, cellular trafficking, post-nuclear entry processing, and cargo gene expression [1]. Due to the lack of naturally encoded viral proteins and helper viruses, each step of AAV transduction heavily relied on the exploitation of host cellular machinery [12]. For example, several surface receptors and co-receptors, including HPSG (heparan sulfate proteoglycan) [13], integrin [14], PDGFR [15], EGFR [16], and KIAA0319L (AAVR) [17], were involved in AAV internalization. Post-entry intracellular trafficking of AAVs required the engagement of endocytosis, and both clathrin and caveolin-dependent as well as -independent mechanisms have been identified [18–21]. Transportation between endosomes and the Golgi apparatus also participated in the AAV sorting and transduction [22]. The trans-Golgi network (TGN)-located receptors, such as GPR108, might directly interact with AAV capsid and be found essential for viral trafficking [23]. Once reached the perinuclear regions, AAV virions then relied on host nuclear import machinery, such as the nuclear pore complex (NPCs) [24,25], for translocation into the nucleus. While in the nucleus, the cargo gene expression was also controlled by the host factors, such as RNF121 [26].

On the other hand, the lack of intrinsic viral factors also made AAV particularly vulnerable to the host anti-viral response. The AAV capsid as well as the unique inverted terminal repeats (ITRs) flanking the transgene expression cassette could serve as a source to activate innate immune responses in human cells [27]. In addition, proteins from diverse cellular pathways were also found to act as host-restricting factors. For example, FKBP52, a cellular chaperon, could bind with the D-sequence in the ITR of the viral genome and inhibit its second-strand

synthesis [28]. MRE11/RAD50/NBS1 complex corresponding to DNA damage response inhibited both ssAAV and scAAV transduction [29,30]. More recently, NP220 and the HUSH complex were reported to silence the AAV genome epigenetically [31]. These findings indicated that our understanding of the mechanism for post-entry AAV transgene expression and the host-AAV interaction was still far from complete. As most of the AAV genomes in the nucleus did not contribute to the transgene expression [32–34], further investigation on host restricting factors and the underlying molecular mechanism might shed light on the development of next generation of gene therapy strategies that could boost AAV transgene expression and minimizing AAV-mediated complications.

In this study, through a genome-wide CRISPR screen, we identified *SMCHD1*, an epigenetic modifier, as a critical broad-spectrum restricting host factor for AAV transduction. *SMCHD1* knock-down by RNAi and CRISPRi or knock-out by CRISPR all resulted in significantly enhanced transgene expression across multiple viral serotypes as well as for both single-strand and self-complementary AAV genome types. Mechanistically, upon viral transduction, SMCHD1 effectively repressed AAV transcription by forming a protein complex with LRIF1, which directly binds with HP1 and the AAV genome to maintain the superstructure of the heterochromatin-like viral genome. *SMCHD1*-KO or *LRIF1*-KD could disrupt such complex and thus enhance accessibilities of AAV DNA and result in transcriptional activation. Together, our results highlight the host factor-induced chromatin remodeling as a critical inhibitory mechanism for AAV transduction and may serve as a target to further improve AAV-based gene therapies.

## Results

### Genome-wide CRISPR-Cas9 screenings identified host restriction factors for AAV transduction

To identify potential host factors involved in AAV transduction, we performed genome-wide CRISPR-Cas9 screenings using the previously reported human sgRNA library "Brunello", which contains 77441 sgRNAs targeting 19114 human genes [35]. The integrity and coverage of the sgRNA library after amplification were first evaluated and confirmed by NGS (**S1 Fig**). To screen for putative restriction factors that inhibit AAV-mediated gene delivery, SpCas9-expressing HeLa cells were transduced with the amplified "Brunello" library by lentivirus with a coverage of 500 cells/sgRNA to generate a mutant cell pool (**Fig 1A**). Two parallel screens were then performed in the mutant cell pool using two scAAV2 vectors, scAAV2-CB-EGFP and scAAV2-EF1-EGFP, which could express EGFP in host cells under the control of chicken beta-globin and EF1a-core promoters respectively to rule out potential regulators on the promoter activity rather than AAV transduction. Both EGFP+ and EGFP- populations were sorted out at 48hrs post-transduction (MOI = 1000 vgs/cell) (**Figs 1A, S2A and S2B**). EGFP-population was defined as the control group while EGFP+ one was defined as the treatment group in the subsequent analysis. Sequencing for sgRNA barcodes in bulk genomic DNA revealed that the vast majority of sgRNAs in the "Brunello" library were represented in the sorted populations, indicating that the sgRNA library was successfully delivered into cells, and almost all the targeting genes went through the screening process (**S2C Fig**).

To evaluate the robustness and effectiveness of our screening process, we first focused on hits that represented genes essential for AAV2 transduction (the percentage of their sgRNAs would decrease in the EGFP+ group). 106 hits were identified in the scAAV2-CB-EGFP screening, while 148 were identified in the scAAV2-EF1-EGFP screening, among which 21 hits outstanding in both screenings were defined as the candidate essential factors for AAV2 transduction (**S2D–S2F Fig**). Among the list of genes, many were also identified and

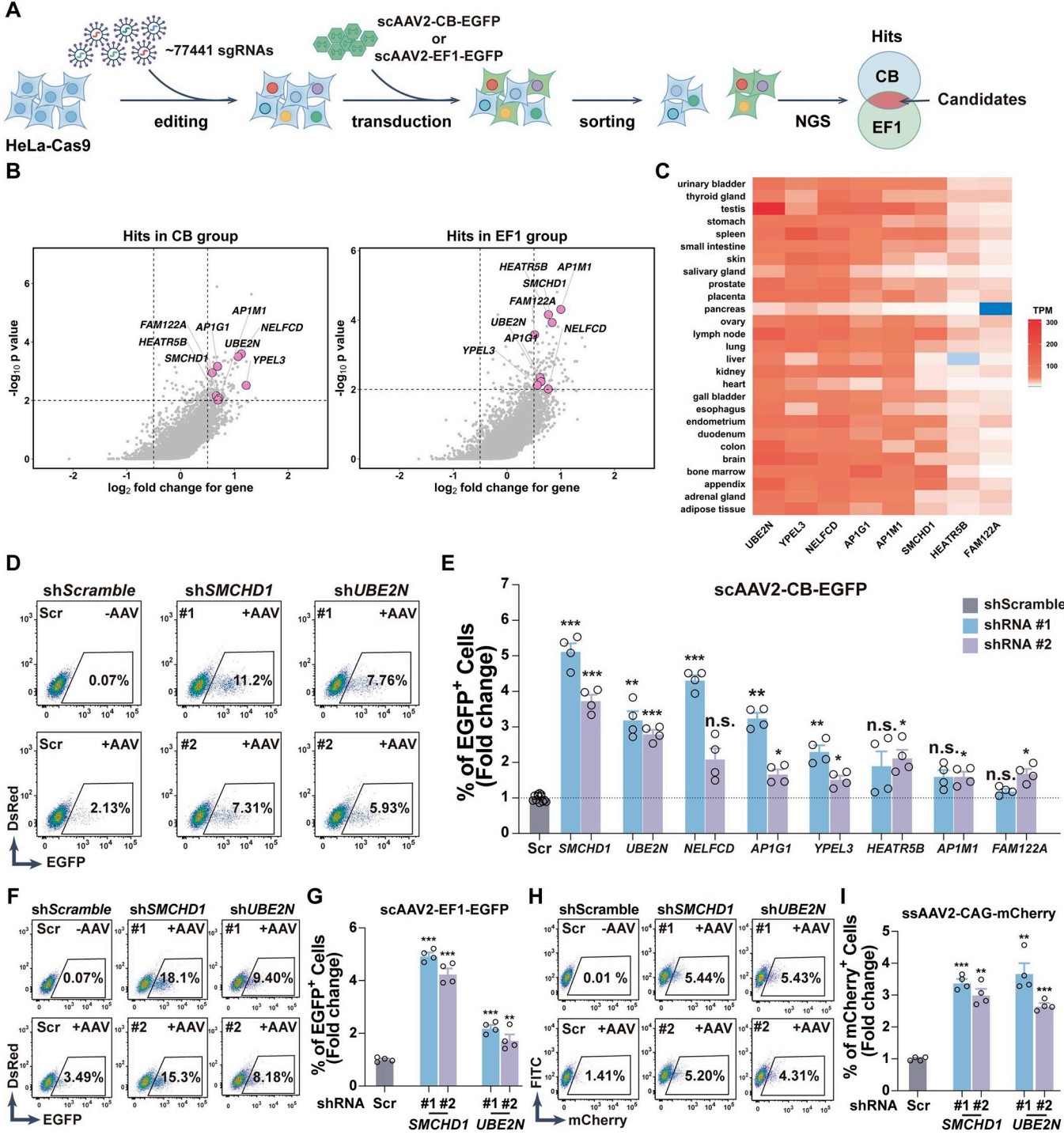

**Fig 1. A genome-wide screen identified candidate host restricting factors for AAV transduction.** (A) Schematic of a genome-wide screen for AAV transduction associated genes. The genome-wide CRISPR library Brunello was delivered into HeLa cells stably expressing SpCas9. scAAV2-CB-EGFP and scAAV2-EF1-EGFP AAVs were then used to transduce gene edited cells. 48hrs post transduction, the cells were fixed and sorted based on EGFP expression. The enrichment of each gRNA in sorted cells was analyzed by MAGeCK. Candidates were defined as the hits only if outstanding in both scAAV2-CB-EGFP and scAAV2-EF1-EGFP screenings. **(B)** Hits of restriction factors for AAV2 transduction were identified. *SMCHD1*, *UBE2N*, *NELFCD*, *AP1G1*, *YPEL3*, *HEATR5B*, *AP1M1*, and *FAM122A* were identified from both screens as candidate restriction factors. **(C)** Candidate genes showed a universal expressing profile across human tissues. RNA-seq data from HPA (human protein atlas) database was normalized using TPM (transcript per million) method. Genes with TPM values ranging from 0–3 were defined as non-expressed genes, indicating by a gradient from blue to white. Genes with TPM values higher than 3 were defined as active genes, with expression levels indicated by a gradient from white to red, representing low to high expression. **(D, E)** FACS analysis indicated

that *SMCHD1* and *UBE2N* knockdown by RNAi significantly enhanced scAAV2-CB-EGFP transduction. Two lentiviral shRNAs for each target gene were designed and transduced into HeLa cells. Scramble shRNA was used as the control. Percentage of EGFP+ cells was quantified by flow cytometry. Non-transduced cells were used as the negative control. Representative FACS data of *Scramble*-KD, *SMCHD1*-KD, and *UBE2N*-KD were shown. Error bar represented data from two independent experiments with duplicate samples. Statistics: One-way ANOVA by SPSS v29.0, ***p < 0.001, **p<0.01, *p<0.05, n. s.: not significant. **(F, G)** *SMCHD1*-KD or *UBE2N*-KD significantly enhanced scAAV2-EF1-EGFP transduction. Representative FACS data of Scramble control, *SMCHD1*-KD, and *UBE2N*-KD were shown. Error bar represented data from two independent experiments with duplicate samples. Statistics: One-way ANOVA by SPSS v29.0, ***p<0.001, **p<0.01. **(H, I)** *SMCHD1*-KD or *UBE2N*-KD significantly enhanced ssAAV2-CAG-mCherry transduction. Representative FACS data of Scramble control, *SMCHD1*-KD, and *UBE2N*-KD were shown. Error bar represented data from two independent experiments with duplicate samples. Statistics: One-way ANOVA by SPSS v29.0, ***p<0.001, **p<0.01.

functionally validated in previous screenings, including *RGP1*, *VPS29*, *VPS52*, *EXT1*, *GPR108*, *TM9SF2*, *NDST1*, and *EXT2* [17,23,36]. These data indicated that our screening and analysis process was robust and effective in uncovering host factors for AAV transduction.

We next sought to investigate the potential host restriction factors for AAV transduction (the percentage of their sgRNAs would increase in the EGFP+ cells). From the screening profile, 106 and 119 hits were identified from the scAAV2-CB-EGFP and scAAV2-EF1-EGFP screenings respectively. 8 overlapping hits were defined as the candidate restriction factors (**Figs 1B and S2G**). Tissue expression profile analysis of these candidate genes using data from the HPA (human protein atlas) database [37] indicated that most of them showed a broad expression distribution among different tissues (**Fig 1C**), suggesting that these potential genes might play a general role in mediating AAV transduction.

## SMCHD1 is a broad-spectrum restriction factor for AAV transduction

To investigate the role of candidate hits in AAV transduction, we used the short-hairpin RNAs (shRNAs) based on RNAi method to knockdown (KD) each candidate gene in HeLa cells and tested the transduction efficiency of scAAV2-CB-EGFP. Two shRNAs per target gene were designed and their KD efficiency was validated (**S3A Fig**). Among the candidate genes, *SMCHD1* knockdown showed the most significant increase of AAV transduction, although all the candidate genes had at least one shRNA with a statistically significant increase upon target knockdown (**Figs 1D, 1E and S3B**). To exclude the possibility that SMCHD1 might act on the activity of the chicken beta-actin (CB) promoter in the scAAV2-CB-EGFP virus, we used another AAV2 with the EF1a-core promoter to test the efficiency and *SMCHD1*-KD would still increase its transduction (**Fig 1F and 1G**), indicating that the effect of *SMCHD1*-KD was not restricted to a specific promoter. As the second-strand synthesis was reported to be a rate-limiting process during AAV-mediated transgene expression [38], we also tested single-strand AAV2 transduction in *SMCHD1*-KD cells, using a ssAAV2-CAG-mCherry virus which expressed mCherry under the control of the CAG promoter. *SMCHD1*-KD still resulted in increased transduction of ssAAV2-CAG-mCherry virus (**Fig 1H and 1I**), indicating SMCHD1's regulation on AAV transduction was independent of dsDNA conversion. Interestingly, the knockdown of *UBE2N*, the second-best candidate, would also increase the transduction of both scAAV2-EF1-EGFP and ssAAV2-CAG-mCherry viruses (**Fig 1F–1I**).

To rule out the potential off-target effects from shRNAs, we also used CRISPRi-mediated *SMCHD1* transcriptional silencing, and the results supported that *SMCHD1* was indeed a host restriction factor for AAV transduction (**Fig 2A–2C**). To further confirm the function of SMCHD1 in AAV transduction, *SMCHD1* knockout cells were also generated by CRISPR/Cas9 (ko*SMCHD1*), and expression of SMCHD1 was validated by Western blotting (**Fig 2D**). *SMCHD1*-KO also resulted in a significant increase of AAV transduction (**Fig 2E and 2F**). Meanwhile, re-introducing *SMCHD1* into KO cells would rescue its suppressive effects on AAV transduction (**Fig 2G and 2H**), indicating the effects were SMCHD1-specific. To

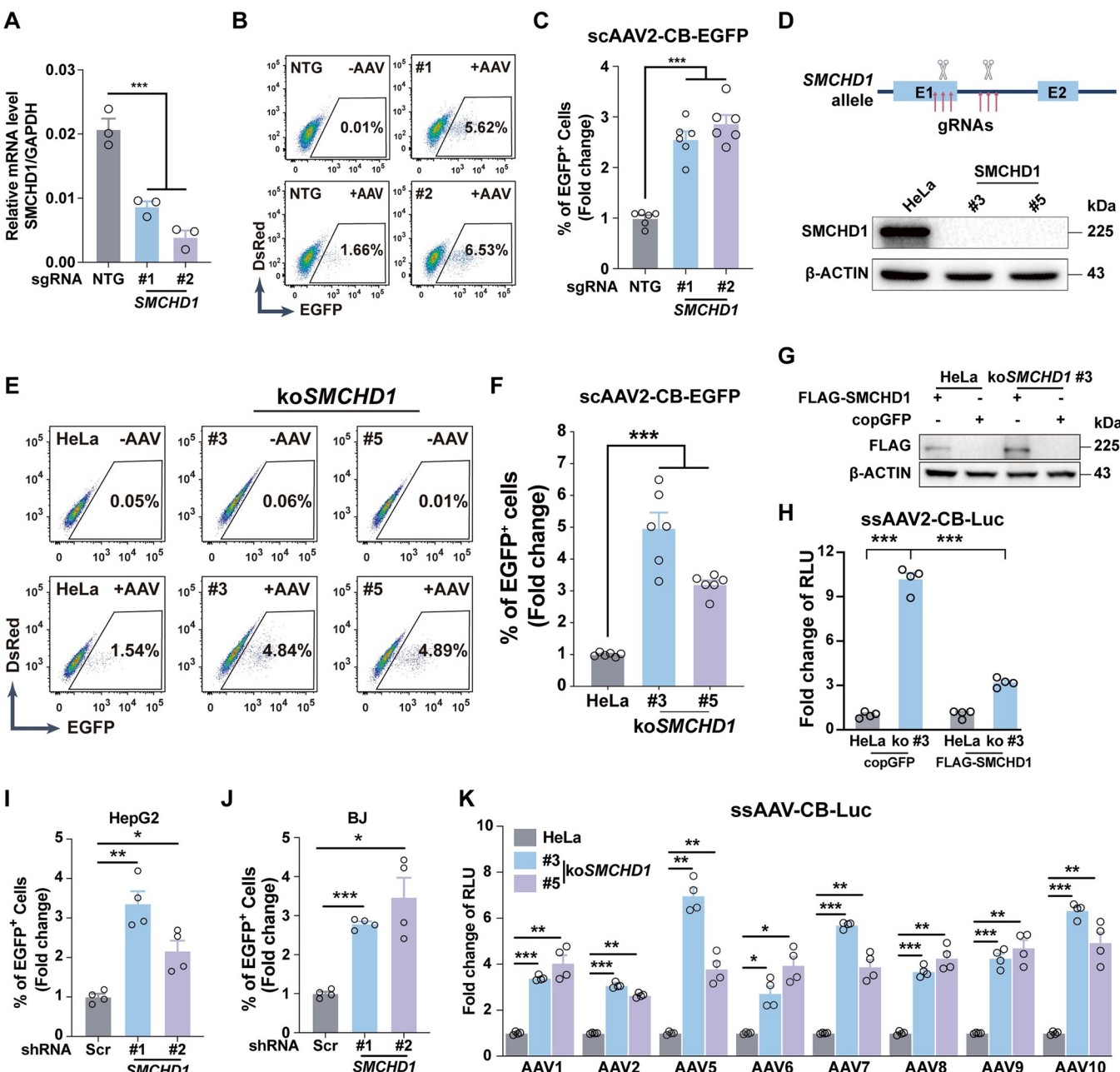

**Fig 2. *SMCHD1* was a broad-spectrum restriction factor for AAV transduction. (A)** RT-qPCR confirmed knockdown of *SMCHD1* by CRISPRi. Two sgRNAs were designed to target within -50 to +200bp region near the *SMCHD1* TSS (transcription starting site). Non-targeting sgRNA (NTG) was used as the control. Error bar represented data from three independent experiments. Statistics: one-way ANOVA by SPSS v29.0, ***p<0.001. **(B, C)** *SMCHD1*-KD by CRISPRi significantly enhanced scAAV2-CB-EGFP transduction in Hela cells. Non-transduced NTG-KD cells were used as the negative control for FACS analysis. Representative FACS data of NTG and *SMCHD1*-KD were shown. Error bar represented data from three independent experiments with duplicate samples. Statistics: One-way ANOVA by SPSS v29.0. ***p<0.001. **(D)** Generation of *SMCHD1*-KO cell lines by CRISPR/Cas9. The top panel showed the KO strategy. Two groups of sgRNA were designed, each containing three sgRNAs to target Exon 1 or Intron 1 of *SMCHD1* respectively. sgRNA vectors were co-transfected into HeLa cells to generate KO cells. The bottom panel showed two *SMCHD1*-KO clones detected by Western blotting. β-ACTIN was used as the loading control. **(E, F)** *SMCHD1*-KO significantly enhanced AAV2 transduction. scAAV2-CB-EGFP AAVs were used for transduction and percentage of EGFP+ cells were quantified by flow cytometry. Non-transduced HeLa cells served as the negative control. Representative FACS data of wt control and *SMCHD1*-KO clones were shown. Error bar represented data from three independent experiments with duplicate samples. Statistics: One-way ANOVA by SPSS v29.0. ***p<0.001. **(G)** FLAG-SMCHD1 was re-introduced into wt HeLa or *SMCHD1*-KO cells by transient transfection. Expression of FLAG-SMCHD1 were detected by Western blotting 24hrs post transfection. β-ACTIN was used as the loading control. **(H)** SMCHD1 complementation reversed the transduction enhancement induced by *SMCHD1*-KO. ssAAV2 expressing luciferase was transduced into cells 24hrs post transfection. RLU (relative light unit) was quantified by microplate reader. Error bar represented data from two independent experiments with duplicate samples. Statistics: *t*-test by SPSS v29.0.

***p<0.001. **(I)** *SMCHD1*-KD significantly enhanced AAV2 transduction in HepG2 cells. Percentage of EGFP+ cells transduced with scAAV2-CB-EGFP were quantified by flow cytometry. Non-transduced cells were used as the gating control. Scramble shRNA was used as the control. Error bar represented data from two independent experiments with duplicate samples. Statistics: One-way ANOVA by SPSS v29.0. **p<0.01, *p<0.05. **(J)** *SMCHD1*-KD significantly enhanced AAV2 transduction in primary human fibroblasts. Error bar represented data from two independent experiments with duplicate samples. Statistics: One-way ANOVA by SPSS v29.0. ***p<0.001, *p<0.05. **(K)** *SMCHD1*-KO enhanced AAV transduction of a broad-spectrum serotypes. Wt Hela cells and *SMCHD1*-KO cells (#3 & #5) were transduced with ssAAV expressing luciferase of AAV1, AAV2, AAV5, AAV6, AAV7, AAV8, AAV9, AAV10. RLU was quantified by microplate reader. Error bar represented data from two independent experiments with duplicate samples. Statistics: One-way ANOVA by SPSS v29.0. ***p<0.001, **p<0.01, *p<0.05.

investigate whether *SMCHD1* would inhibit AAV2 transduction in other cells, gene knock-down by shRNAs was performed in both HepG2 and primary human skin fibroblasts (BJ cells) (**S4A and S4C Fig**) and *SMCHD1*-KD significantly increased AAV transduction in both cells (**Figs 2I, 2J, S4B and S4D**). To determine whether *SMCHD1* would act as a general restriction factor for AAVs, we transduced *SMCHD1*-KO cells with a panel of luciferase-expressing AAVs from different serotypes, including AAV1, 2, 5, 6, 7, 8, 9, and 10, and discovered that *SMCHD1*-KO significantly increased viral transductions for all the tested serotypes (**Fig 2K**). Together, these data demonstrated that *SMCHD1* is a general and broad-spectrum host restriction factor to repress AAV transduction.

## SMCHD1 directly bound the AAV genome to repress its transcription

The successful AAV transduction relies on multiple steps and molecule interactions, including virions binding with the host cell membrane, internalizing and trafficking to TGN, nucleic entry, dsDNA conversion, and transgene expression, during which many host proteins were recruited and utilized to facilitate viral transduction [12]. To dissect the mechanism of *SMCHD1*'s restricting role in AAV transduction, we first tested whether *SMCHD1* knockout would contribute to the permissivity of host cells to AAV virions. Quantification of the total copy number of viral DNAs within the transduced cells showed no difference among wt- and *SMCHD1*-KO cells (**Fig 3A**). In addition, further quantification of viral genomes in mem-brane-bound, cellular uptake, and nucleic entry, showed no significant change after *SMCHD1* depletion either (**Fig 3A and 3B**), suggesting that virion entry and trafficking to the nucleus were not affected by *SMCHD1* knockout. On the other hand, mRNA transcriptional level was substantially increased in *SMCHD1*-KO cells compared with wt HeLa cells when they were transduced with the ssAAV2-CB-Luc virus at different MOIs (**Fig 3C**). These data suggested that *SMCHD1* did not regulate the cellular trafficking of AAV particles but functioned at the transcriptional level to repress AAV transgene expression.

Originally identified as a key regulator in X chromosome inactivation [39] and mainte-nance of super-structure formation in the inactive X [40], *SMCHD1* has been known to have a role in maintaining genome organization. *SMCHD1* mutations were also identified as a causal factor in two distinct human disorders, Facioscapulohumeral muscular dystrophy (FSHD) and Bosma arhinia and microphthalmia (BAMS), both of which showed aberrations in genomic structure and epigenetic modifications [41]. Based on these findings, we hypothesized that SMCHD1 might directly interact with AAV genomic DNA after virion uncoating and cause transcriptional repression. Indeed, when AAV virions were present, immunoprecipitation of FLAG-tagged SMCHD1 could pull down AAV genomic DNAs (**Fig 3D and 3E**). Immuno-staining of endogenous SMCHD1 and DNAscope staining of AAV genomic DNA further con-firmed the co-localization of these two components (**Figs 3F, 3G and S5A**). Similar results could also be observed in human BJ fibroblasts (**S5B and S5C Fig**). Together, these results indicated that SMCHD1 was probably not involved in the initial virion binding, cellular uptake, and intracellular trafficking, but instead would directly interact with the AAV viral

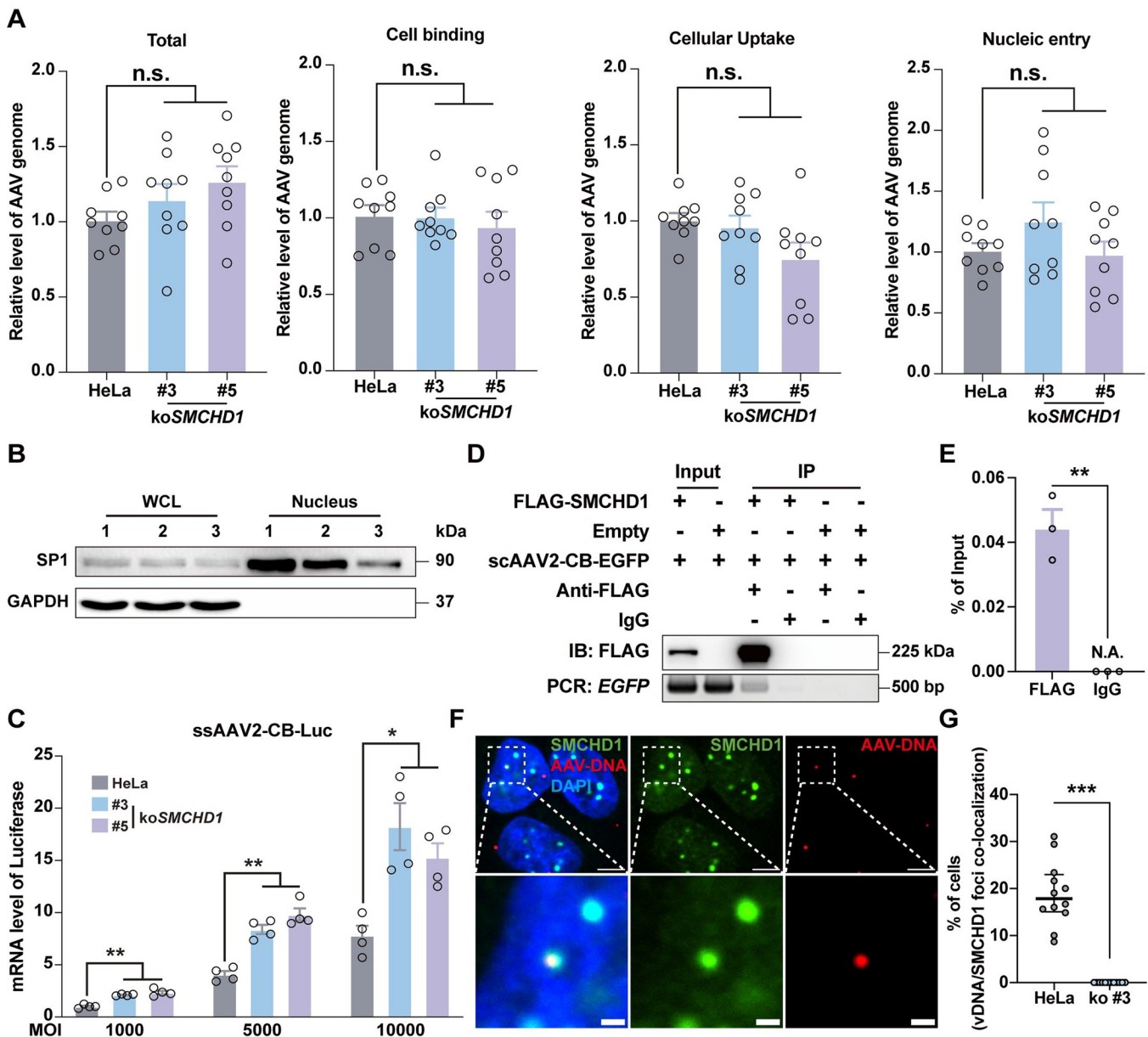

**Fig 3. SMCHD1 directly bound AAV genome and repressed its transcription. (A)** *SMCHD1*-KO did not affect AAV binding, uptake, and nucleic entry. Total copies of AAV genomic DNA or those bound with cell membrane or intracellular uptake copies, or intra-nucleic copies were quantified respectively post transduction of scAAV2-CB-EGFP (MOI: 1000 vgs/cell). Error bar represented data from three independent experiments with triplicate samples. Statistics: One-way ANOVA by SPSS v29.0. n.s.: not significant. **(B)** Isolation of cell nucleus was validated by Western blotting. WCL, whole cell lysate. SP1, marker of nucleic protein. GAPDH, marker of cytoplasmic protein. **(C)** *SMCHD1*-KO significantly promoted AAV RNA transcription. Total RNAs were extracted at 48hrs post transduction with ssAAV2-CB-Luc (MOI: 1000, 5000, 10000 vgs/cell, respectively). mRNA levels of luciferase were quantified by RT-qPCR. Error bar represented data from two independent experiments with duplicate samples. Statistics: One-way ANOVA by SPSS v29.0. **p<0.01, *p<0.05. **(D, E)** SMCHD1 directly bound AAV genomic DNAs. FLAG-SMCHD1 expressing or empty plasmids were transfected in HEK293 cells. scAAV2-CB-EGFP were transduced in cells at 48hrs post transfection. Cells were collected for immunoprecipitation using anti-FLAG antibody or normal IgG at 48hrs post transduction. FLAG-SMCHD1 was detected using Western Blotting. The presence of AAV-DNAs was confirmed in the IPed samples by PCR and quantified by qPCR. Error bar represented data from three independent experiments. Statistics: *t-test* by SPSS v29.0. **p<0.01. **(F, G)** Co-localization of endogenous SMCHD1 with AAV DNAs in nucleus. AAV genome DNAs were detected using the probe for the anti-sense strand of luciferase by DNAscope. DAPI was used for nuclear staining. Scale bars: 5μm, 1μm (zoom-in). Percentage of cells with vDNAs and SMCHD1 co-localization signals were quantified in **(G)**. Error bar represented data from 12 randoms views of two independent experiments. Statistics: t-test by SPSS v29.0, ***p<0.001.

genomic DNAs in the nucleus, which then resulted in transcriptional repression of the viral genome.

## The SMCHD1-LRFI1 complex repressed AAV transcription

As mentioned above, SMCHD1 was previously reported to have a pivotal role in maintaining X chromosome inactivation by facilitating super-structure formation on the inactive X [39,42]. LRIF1 (ligand-dependent nuclear receptor-interacting factor 1) was responsible for loading SMCHD1 on the target genomic regions and compacting Xi in a manner dependent on HP1 (heterochromatin protein 1) [43,44]. Moreover, both SMCHD1 and LRIF1 were found to be telomere-binding proteins [45], suggesting that the SMCHD1-LRIF1 complex may have cellular functions that extended from the original X chromosome inactivation. These findings prompted us to test the hypothesis that the SMCHD1-LRIF1 complex might play a role in transcriptional repression of the AAV genome.

We first confirmed the co-localization of SMCHD1 and LRIF1 in HeLa cells (**Fig 4A**). To investigate whether LRIF1 is involved in regulating AAV transcription, we performed shRNA-mediated knockdown of *LRIF1* (**Fig 4B**). *LRIF1*-KD significantly enhanced both scAAV2 and ssAAV2 transcription, although the effects were not as pronounced as *SMCHD1*-KD (**Fig 4C and 4E**). To exclude the potential off-target effects from shRNAs, a rescue experiment using FLAG-tagged LRIF1 was carried out and supplementing the shRNA-resistant LRIF1 would abolish the AAV transcription enhancement in *LRIF1*-KD cells (**Fig 4D and 4E**).To examine if the AAV transcription enhancement from *LRIF1*-KD cells was dependent on SMCHD1, we further knocked down *LRIF1* in *SMCHD1*-KO cell lines with shRNAs (**Fig 4F**) and indeed, the enhancement was largely abolished in *SMCHD1*-KO cells (**Fig 4G**). Together, these data suggested that SMCHD1 also formed a complex with LRIF1 in HeLa cells and knockdown of LRIF1 would also result in SMCHD1-dependent increase of AAV transcription.

## The SMCHD1-LRIF1-HP1 complex maintained AAV genome in a heterochromatin-like state

The AAV genome was known to be associated with histones and form a chromatin-like structure in transduced cells [46]. While during X chromosome inactivation, the SMCHD1-LRIF1 complex would directly bind a heterochromatin protein, HP1, to initiate chromatin compaction and heterochromatin formation [43], possibly through HP1-mediated phase separation [47]. These findings prompted us to hypothesize that a similar mechanism may be involved in the SMCHD1-LRIF1 complex-mediated suppression of the AAV genome.

To test the hypothesis, immunostaining of HP1, LRIF1, and SMCHD1 was performed in HeLa cells and the three indeed co-localized with each other in the nucleus (**Fig 5A**). The co-localizations between HP1 vs SMCHD1 or LRIF1 were significantly reduced upon *SMCHD1*-KO or *LRIF1*-KD (**Fig 5A**). To investigate if the interaction of HP1 and SMCHD1 with LRIF1 was essential for their suppressive role in AAV transduction, two LRIF1 mutants which lack SMCHD1-binding (m1) or both HP1 and SMCHD1-binding domains (m2) were constructed [43] (**Fig 5B**). Indeed, removing the respective binding domains would cause specific loss of interaction between LRIF, SMCHD1 and HP1, as revealed by co-immunoprecipitation (**Fig 5C**). Morever, overexpression of LRIF1 mutants failed to rescue the knockdown effects of sh*LRIF1* on AAV transduction (**Fig 5D and 5E**). Immunoprecipitation of HP1α coupled qPCR analysis (**Fig 5F and 5G**) and co-staining of HP1 and AAV DNAs (**Fig 5H**) further confirmed the direct interaction and co-localization of HP1α with AAV genomic DNAs. When *LRIF1* was knocked down by shRNAs, the co-localization of SMCHD1 and HP1 was disrupted, suggesting that the binding between SMCHD1 and HP1 was LRIF1 dependent (**Fig 5I**).

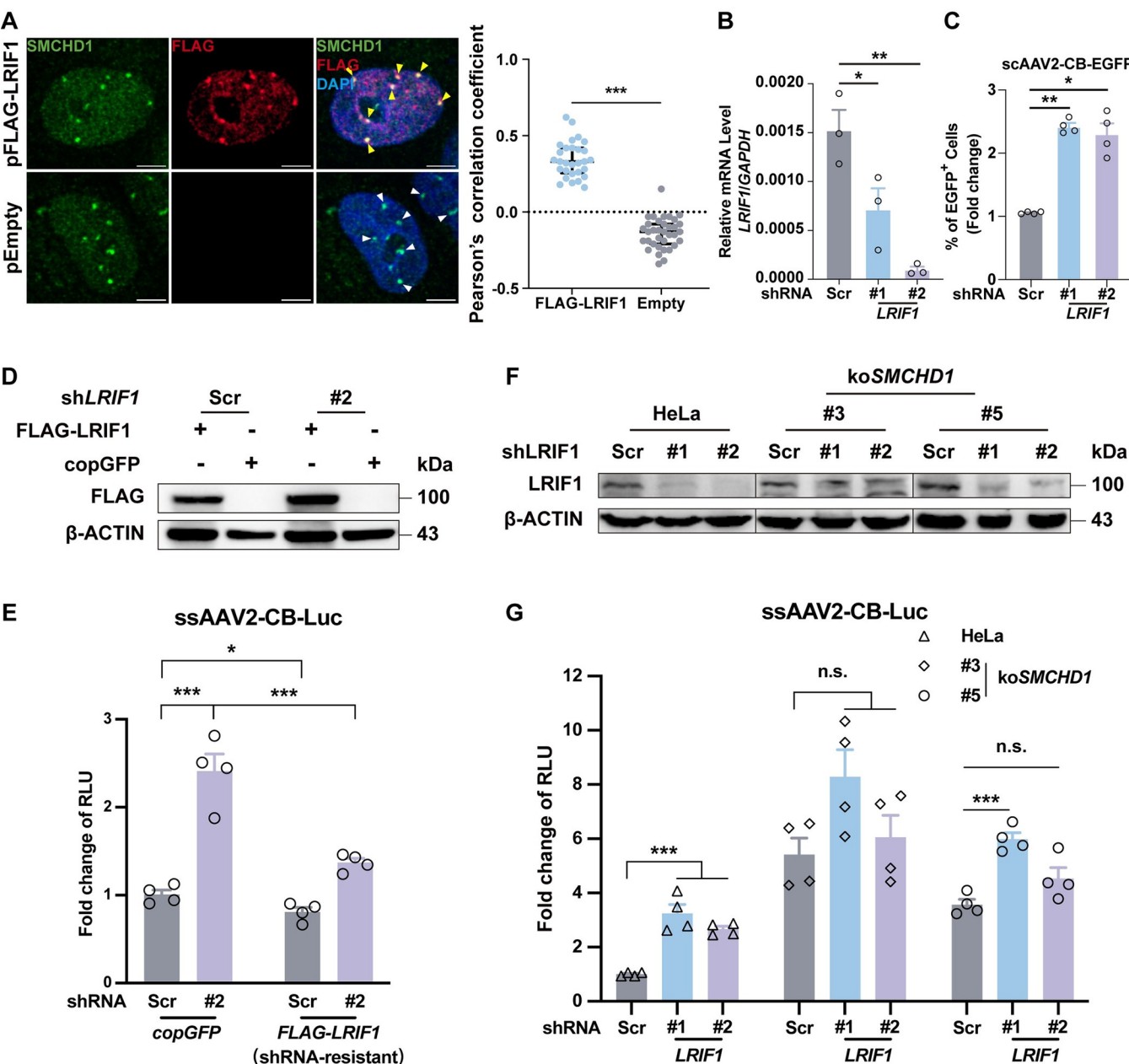

**Fig 4. LRIF1 was involved in SMCHD1-mediated repression on AAV transcription. (A)** SMCHD1 colocalized with LRIF1 in nucleus. Left: FLAG-LRIF1 or empty plasmid were transfected in HeLa cells. Cells were fixed for immunostaining with anti-SMCHD1 and anti-FLAG antibodies at 24hrs post transfection. Scale bar: 5μm. Right: Co-localization was quantified using the PCC (Pearson's correlation coefficient) method by coloc2 plugin in Image J. Each dot represents a single cell. n = 29 (FLAG-LRIF1), 35 (Empty), respectively. Statistics: *t*-test by SPSS v29.0, ***p<0.001. **(B)** RT-qPCR confirmed knockdown of *LRIF1* in HeLa cells. Error bar represented data from three independent experiments. Statistics: One-way ANOVA by SPSS v29.0, *p<0.05, **p<0.01. **(C)** *LRIF1*-KD also enhanced scAAV2 transduction. Percentage of EGFP+ cells were quantified by flow cytometry. Error bar represented data from two independent experiments with duplicate samples. Statistics: One-way ANOVA by SPSS v29.0. *p<0.05, **p<0.01. **(D)** Design of a shRNA-resistant LRIF1 mutant. shRNA targeting region in LRIF1 coding sequence was mutated synonymously. FLAG tagged mutant LRIF1 or copGFP were transduced in *LRIF1*-KD or *Scramble*-KD HeLa cells by lentiviral vectors. Protein expression was detected by Western Blotting with anti-FLAG antibody. β-ACTIN was used as the loading control. **(E)** *LRIF1* complementation reversed the transduction enhancement induced by *LRIF1*-KD. Mutant *LRIF1* or *copGFP* were transduced in *LRIF1*-KD HeLa cells by lentiviral vector. Error bar represented data from two independent experiments with duplicate samples. Statistics: *t*-test by SPSS v29.0. ***p<0.001, *p<0.05. **(F)** LRIF1 was efficiently knock-down in HeLa and *SMCHD1*-KO cell lines by shRNAs. **(G)** Enhancement of AAV2 transduction by *LRIF1*-KD was dependent on SMCHD1. Error bar represented data from two independent experiments with duplicate samples. Statistics: One-way ANOVA by SPSS v29.0. ***p<0.001, n.s.: not significant.

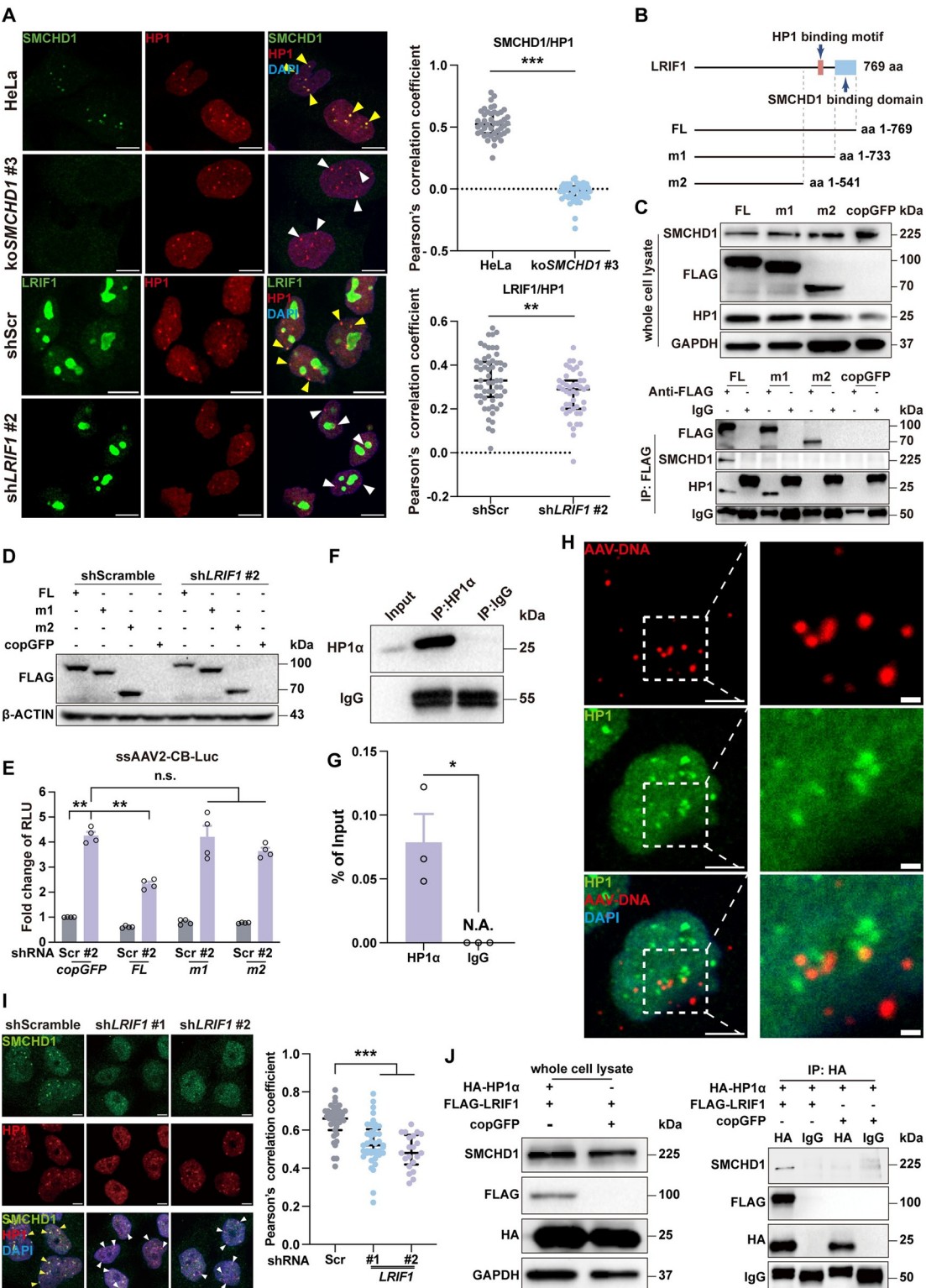

**Fig 5. SMCHD1/LRIF1/HP1 complex repressed AAV transcription. (A)** LRIF1 and SMCHD1 co-localized with HP1, a marker of heterochromatin. Left: Untransduced wt HeLa, *SMCHD1*-KO, Scramble control, and *LRIF1*-KD cells were fixed and immunostained with both anti-SMCHD1 and anti-HP1 or with both anti-LRIF1 and anti-HP1 antibodies. Yellow arrowheads: co-localization of SMCHD1/HP1 or LRIF1/HP1. White arrowheads: HP1 foci without SMCHD1 or LRIF1 co-localization. Scale bar: 10μm. Right: co-localization of SMCHD1/HP1 or LRIF1/HP1 were quantified by PCC method using coloc2 plugin in Image J.

Each dot represents a single cell. n = 46 (HeLa), 43 (koSMCHD1 #3), n = 57 (shScr), and 45 (sh*LRIF1* #2), respectively. Statistics: *t*-test by SPSS v29.0, \*\*\*p<0.001, \*\*p<0.01. (**B**) Design of LRIF1 mutants. A schematic showed the design of LRIF1 mutants by removing SMCHD1-binding domain (m1) or removing both SMCHD1-binding and HP1-binding domain (m2). (**C**) Immunoprecipitation confirmed the interaction among SMCHD1, HP1 and LRIF1 as well as its mutants. Top: expression of LRIF1 mutants and associated proteins were confirmed using whole cell lysates of HEK293 cells by Western blotting at 48hrs post transfection. Bottom: Immunoprecipitation of cells transfected with different LRIF1 vectors by anti-FLAG antibody or IgG. (**D**) Expression of full-length or mutant LRIF1s in Scramble control or *LRIF1*-KD cells were confirmed by Western blotting. β-ACTIN was used as the loading control. (**E**) LRIF1 mutants that lack HP1 and/or SMCHD1 binding failed to repress AAV transduction. ssAAV2 expressing luciferase was transduced in cells and RLU was detected by microplate reader. Statistics: One-way ANOVA by SPSS v29.0. \*\*p<0.01, n.s.: not significant. (**F**) Confirmation of HP1α immunoprecipitation. Lysates of HeLa cells transduced with scAAV2-CB-EGFP were used for immunoprecipitation by anti-HP1A antibody or IgG. (**G**) HP1α directly bound AAV genome. Level of co-immunoprecipitated AAV DNAs were quantified by qPCR. Error bar represented data from three independent experiments with single sample. Statistics: *t*-test by SPSS v29.0. \*p<0.05. (**H**) Co-localization of HP1 with AAV DNAs in HeLa cells. Scale bars: 5μm and 1μm (zoom-in). (**I**) *LRIF1*-KD disrupted SMCHD1/HP1 colocalization. *LRIF1*-KD cells were fixed for immunostaining of SMCHD1 and HP1, Scramble control (Scr) cells were used as the control. Scale bar: 5μm. Co-localization was quantified using PCC method by coloc2 plugin in Image J. Each dot represents a single cell. n = 35 (shScramble), 40 (sh*LRIF1* #1), 21 (sh*LRIF1* #2), respectively. Statistics: One-way ANOVA by SPSS v29.0. \*\*\*p<0.001. (**J**) LRIF1 mediated the interaction between SMCHD1 and HP1. Left: expression of FLAG-tagged LRIF1 and HA-tagged HP1α were confirmed by Western blotting using the whole cell lysates of HEK293 at 48hrs post transfection. GAPDH served as the loading control. Right: lysates of HEK293 cells transfected with both HA-HP1α and FLAG-LRIF1 or with both HA-HP1α and copGFP expressing plasmids were used for immunoprecipitation by anti-HA antibody or IgG.

Similarly, endogenous SMCHD1 could only be pulled down by HP1α in the presence of LRIF1 (**Fig 5J**). While in *SMCHD1*-KO cells, the interaction between LRIF1 and HP1 was also disrupted (**S6 Fig**).

In addition, *SMCHD1*-KO or *LRIF1*-KD did not seem to affect the protein expression of HP1 (**Fig 5A and 5I**), immunoprecipitation of HP1α-coupled qPCR and co-staining analysis of HP1 and AAV DNAs also indicated unaffected HP1-AAV DNA association (**Fig 6A–6F**). However, loss of SMCHD1 indeed caused a significant increase of genome accessibility of AAV DNAs as revealed by ATAC-qPCR, while active transcribed and desert loci were not affected (**Fig 6G**). *LRIF1*-KD showed similar effects, although to a lesser extent probably due to the existence of residual LRIF1 protein (**Fig 6H**). Together, these data indicated that HP1 signals on DNA might be upstream of SMCHD1 or LRIF1, and SMCHD1-LRIF1 complex was indeed recruited onto the HP1-bound AAV genome to potentially compact and form the superstructure of the heterochromatin-like viral genome and suppress its transcription. Loss of such suppression would result in increased genome accessibility and thus higher gene transcription.

Meanwhile, *LRIF1* was also reported to inhibit retinoic acid receptors (RARs) mediated ligand-dependent transcriptional activation by recruiting histone deacetylases (HDACs) [48]. To determine whether deacetylation affects AAV transduction, Trichostatin A (TSA), a potent HDAC inhibitor, was used to treat wt- and *SMCHD1*-KO cells. Although levels of acetylated histone showed a dose-dependent increase upon TSA treatment (**Fig 6I**), the transduction of AAV2 showed no difference after TSA treatment in both wt- and *SMCHD1*-KO cells (**Fig 6J**). These data indicated that histone acetylation was unlikely involved in SMCHD1-mediated suppression on AAV transduction.

On the other hand, loss of SMCHD1 was reported to cause DNA hypomethylation and reduced H3K9me3 modifications on the Xi and D4Z4 repeats [39,43,49,50]. Therefore, we also tested the CpG methylation level in the AAV genome. However, only a low level of DNA methylation was detected by bisulfite sequencing in the chicken beta-globin promoter of the AAV viral genome (**S7A Fig**), suggesting that DNA methylation was unlikely to be part of SMCHD1-mediated suppression. Additionally, we also quantified the H3K9me3 and H3K27me3 levels on scAAV2-CB-EGFP in transduced cells by immunoprecipitation. Neither H3K9me3 nor H3K27me3 modification showed any significant difference between wt- and

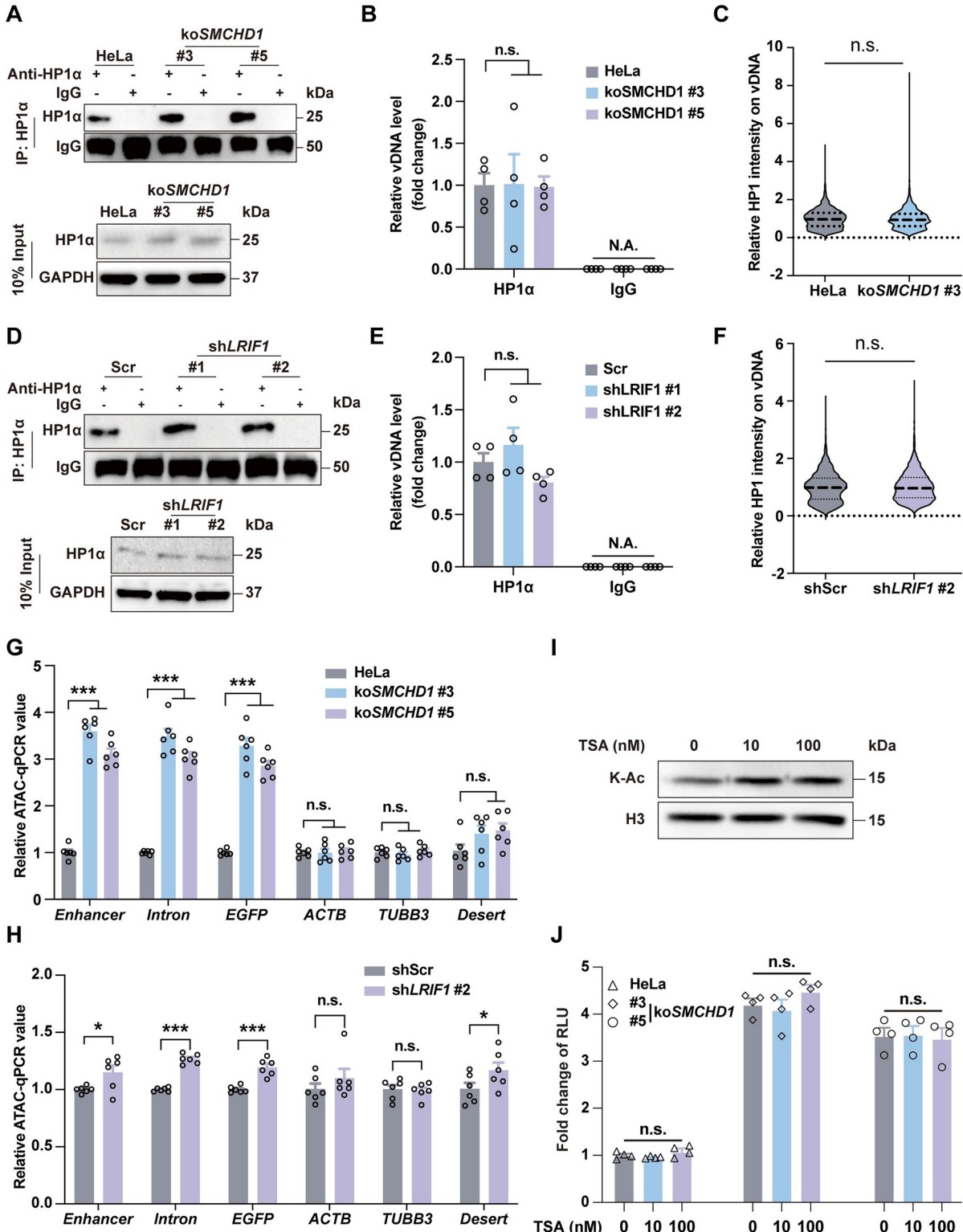

**Fig 6. SMCHD1/LRIF1/HP1 complex repressed AAV transcription through suppressing accessibility of vDNAs. (A)** Confirmation of HP1α immunoprecipitation. Lysates of HeLa-wt or *SMCHD1*-KO cells transduced with scAAV2-CB-EGFP were used for immunoprecipitation by anti-HP1α antibody or IgG. **(B)** *SMCHD1*-KO did not affect the HP1α-bound vDNA level. Level of co-immunoprecipitated AAV DNAs were quantified by qPCR. Error bar represented data from two independent experiments with duplicate samples. Statistics: One-way ANOVA by SPSS v29.0, n.s.: not significant. **(C)** *SMCHD1*-KO did not affect HP1 intensity on vDNAs. HP1

intensities on vDNAs in nuclear were quantified using the mean gray value of signals by Image J. Data were from eight views of two independent experiments. Statistics: *t*-test by SPSS v29.0. n.s.: not significant. **(D)** Confirmation of HP1α immunoprecipitation. Lysates of Scramble control or *LRIF1-KD* cells transduced with scAAV2-CB-EGFP were used for immunoprecipitation by anti-HP1A antibody or IgG. **(E)** *LRIF1*-KD did not affect the HP1α-bound vDNA level. Error bar represented data from two independent experiments with duplicate samples. Statistics: One-way ANOVA by SPSS v29.0, n.s.: not significant. **(F)** *LRIF1*-KD did not affect HP1 intensity on vDNAs. Data were from eight views of two independent experiments. Statistics: *t*-test by SPSS v29.0. n.s.: not significant. **(G)** *SMCHD*-KO increased genome accessibility of AAV DNAs. Accessibilities of scAAV2-CB-EGFP DNAs in wt HeLa and *SMCHD1*-KO cells were quantified by ATAC-qPCR. *Enhancer*, *Intron*, and *EGFP* represented the CMV enhancer, SV40 intron, and EGFP gene body regions in AAV genome, respectively. *ACTB*, *TUBB3*, and *Desert* indicated *ACTB*, *TUBB3*, and gene desert region in human genome, respectively. Error bar represented data from two independent experiments with triplicate samples. Statistics: One-way ANOVA by SPSS v29.0. ***$p < 0.001$, n.s.: not significant. **(H)** *LRIF1*-KD improved accessibilities of AAV genome DNAs. Error bar represented data from two independent experiments with triplicate samples. Statistics: *t*-test by SPSS v29.0. ***$p < 0.001$, *$p < 0.05$, n.s.: not significant. **(I)** Trichostatin A (TSA) treatment significantly increased histone acetylation. Acetylated histone was probed by anti-pan acetylated lysine antibody using Western blotting after TSA treatment (0, 10, and 100nM) for 48hrs. H3 was used as the loading control. **(J)** TSA treatment had no effects on AAV transduction. wt and *SMCHD1*-KO HeLa cells were transduced with ssAAV2-CB-Luc (MOI: 1000 vgs/cells) and treated with TSA (0, 10, and 100nM) for 48hrs. Error bar represented data from two independent experiments with duplicate samples. Statistics: One-way ANOVA by SPSS v29.0, n.s.: not significant.

*SMCHD1*-KO cells, and there was no significant change in the overall modifications in the nucleus either (**S7B and S7C Fig**). Therefore, we concluded that SMCHD1 does not affect the DNA methylation and histone modifications on the AAV genome.

## Discussion

In this study, we have performed an unbiased genome-wide CRISPR screening and identified restriction host factors for AAV transduction. Eight hits outstanding in both scAAV2-CB-EGFP and scAAV2-EF1-EGFP screenings were defined as putative restriction factors suppressing AAV-mediated transgene delivery in the host cells. Among these hits, we identified that *SMCHD1* was a broad-spectrum restriction factor of AAV infection, depletion of which could enhance AAV transduction of multiple serotypes, and was independent of ssAAV and scAAV genotypes. Furthermore, we demonstrated that expression of the cargo gene on the AAV vector was effectively repressed transcriptionally by the binding between SMCHD1 and the viral genome. Such suppression is probably due to SMCHD1/LRIF1 complex-mediated compaction of the viral genomic DNA and reduction of viral chromatin accessibility (**Fig 7**). Together, these results highlighted that the process of chromatin remodeling might be a critical host pathway restricting AAV transduction.

It has long been known that most of the AAVs do not contribute to transgene expression post nucleic entry [32–34]. Much effort has thus been devoted to unraveling host factors that impede AAV transduction. Several host pathways have been proposed to restrict AAV transduction, including SUMOylation [51,52], DNA damage response [30,53], RNA processing [54], and epigenetic regulation [31,55–57], while the detailed regulatory mechanisms are still largely incomplete. From our study, the SMCHD1-LRIF1 complex was identified as a critical regulator of AAV genome accessibility and transcription. SMCHD1 was originally characterized to function in X chromosome inactivation by recruiting LRIF1 and HP1 to compact and maintain the formation of heterochromatin superstructure of the inactive X [42,43]. The discovery here extended the role of the SMCHD1-LRIF1 complex to a cellular anti-viral mechanism that restricted AAV transduction. Interestingly, this anti-viral role of *SMCHD1* may very likely be a general mechanism, as a recent study also identified it as a host restriction factor for the replication of herpesviruses [58]. Therefore, harnessing the restricting role of *SMCHD1* in AAV transduction may provide a novel target to improve the cargo delivery efficiency of current gene therapy.

Meanwhile, it was recently reported that epigenetic modifications might play a critical role in regulating AAV transduction [31,56]. One study revealed that AAV capsid protein might be

**Fig 7. SMCHD1/LRIF1/HP1 complex repressed AAV transcription through maintaining a heterochromatin-like state.** Schematic of SMCHD1 as a host restriction factor for AAV transduction.

involved in driving the active epigenetic marking (H3K4me3 and H3K27ac) of AAV genomes, which strongly influenced the expression of cargo genes [56]. It was also reported that the DNA binding protein NP220, in association with the human silencing hub (HUSH) complex, could suppress both single-strand and self-complementary AAVs [31]. NP220 KO would result in increased AAV transcription and be correlated with decreased H3K9me3 modifications in the AAV genomes [31]. In our study, we did not detect any significant changes in H3K9me3 modifications on the AAV genome, although the modification itself was indeed present in the viral genome. Neither any CpG methylation detected in the promoter region of AAV cargo genes nor modulation of acetylation by HDAC inhibitors would have any impact on *SMCHD1*-mediated suppression of AAV transduction. These data suggested that the *SMCHD1*-mediated cellular mechanism might be independent of epigenetic regulation on H3K9me3 and DNA methylation. Further studies on the detailed molecular mechanism may yield more insights into the SMCHD1's role in host-pathogen interactions.

## Materials and methods

### Cell culture

HeLa, HeLa derived SMCHD1-KO cells, 293FT (Invitrogen, R7007), and HEK293 cells (ATCC, CRL-1573) were cultured in Dulbecco's modified Eagle's medium (DMEM) (Gibco, C11995500BT) supplemented with 10% fetal bovine serum (FBS) (PAN-Biotech, ST30-3302)

and antibiotics (penicillin 100U/mL, Streptomycin 100 μg/mL, Amphotericin B 250 ng/mL) (Solarbio, P7630) and passaged at 80–90% confluence using 0.25% Trypsin-EDTA (Gibco, 25200072). Human foreskin fibroblast BJ (ATCC, CRL-2522) and HepG2 (MeisenCTCC, CTCC-001-0014) were cultured in DMEM supplemented with 10% FBS and antibiotics (penicillin 100 U/mL, Streptomycin 100 μg/mL, Amphotericin B 250 ng/mL) and passaged at a ratio of 1:4 every three days using 0.25% Trypsin-EDTA. All the cells were maintained under standard culture conditions (37˚C, 5% $CO_2$, humidified atmosphere).

## Cloning and plasmid constructs

pCRISPRi was generated by cloning U6 promoter, EF1a promoter, and Cas9m4 sequence amplified by PCR, and synthesized KRAB-MeCP2-P2A-PuroR sequence (The Beijing Genomics Institute) into a backbone derived from a PB transposon plasmid (gifted from Prof. Chong Liu at Zhejiang University). pPB-CRISPR-copGFP was generated by cloning the sgRNA expression cassette amplified from LentiGuide-Puro (Addgene, 52963), EF1a promoter, Cas9-P2A segment amplified from LentiCas9-Blast (Addgene, 52962), and copGFP segment amplified from pCDH-EF1-copGFP (Addgene, 73030) into the PB transposon backbone plasmid. pPB-CRISPR-mCherry was generated by cloning sgRNA expression cassette, EF1a promoter, Cas9-P2A segment, and mCherry segment into the PB transposon backbone plasmid. pPB-EF1-BSD was generated by cloning EF1a promoter, SV40-pA segment, PGK promoter, and BSD segment into the PB transposon plasmid. pUC-FLAG-SMCHD1 was generated by cloning 3xFLAG sequence and *SMCHD1* sequence amplified by PCR from HeLa cDNA into pUC19 (Addgene, 50005). FLAG-SMCHD1 segment was then cut and ligated into pPB-EF1-BSD to generate pPB-EF1-FLAG-SMCHD1. pLVX-3xFLAG-BSD was derived from pLVX-BSD (gifted from Dr.Hu Chen at Chengdu Medical College) by inserting 3x FLAG segment adjacent to the CMV promoter at 3' end. *LRIF1* was amplified from cDNA and the shRNA complementary region of LRIF1 was synonymously mutated during amplification (from 5'-TTCAGATGTGTCACAACATAA-3' to 5'-CAGCGACGTGAGCCAGCACAA-3'). pLVX-FLAG-LRIF1-BSD was generated by ligating shRNA-resistant LRIF1 into pLVX-3XFLAG-BSD. LRIF1 mutants (m1 and m2) plasmids were then cloned using site-directed mutagenesis kit (Vazyme, C214) on the base of pLVX-FLAG-LRIF1-BSD. HP1α was amplified from cDNA and HA-coding sequence was added by PCR. pLenti-HA-HP1α-BSD was generated by cloning HA-HP1α segment, PGK-BSD segment into the LentiCas9-Blast backbone. Sequence of all the cloning plasmids was confirmed by Sanger sequencing (Sangon Biotech (Shanghai) Co., Ltd).

## Lentivirus production

293FT cells were seeded on 12-well plates at $3x10^5$ cells/well and cultured overnight. Three plasmids, pMD2.G (a gift from Biao Dong), psPAX (a gift from Biao Dong), and transfer plasmids, were transfected in cells at 1:2:4 ratio. Supernatant virus was harvested 48hrs later, clarified by centrifugation at 4000 rpm for 10 min in an Eppendorf tabletop centrifuge, and then for using immediately or stocking at -80˚C.

## AAV production

Besides ssAAV2-CAG-mCherry was purchased from Genewiz, other AAV vectors were produced using a triple plasmid co-transfection method as previously reported [59]. One vector plasmid, one helper plasmid of indicated serotype, one mini adenovirus function helper plasmid pFΔ6, were co-transfected into HEK293 cells cultured in roller bottles at a ratio of 1:2:1. The transfected cells were harvested 3 days later. rAAVs were then purified by two rounds of

cesium chloride–gradient ultracentrifuge. After extensive buffer exchange against phosphate-buffered saline with 5% D-sorbitol, the peak fractions of purified virus were pooled and stored at −80˚C before used.

## Brunello library amplification and NGS

Brunello library (Addgene, 73178) was amplified using HST08 electrocompetent cells (Takara, 9028) by electroporation. Total ten transforming reactions were set each containing 25μL competent cells plus 10ng plasmids. All the transformant was mixed with 300mL Luria-Bertani (LB) medium (Invitrogen, 12780052) and cultured in a shaker for 12hrs at 37˚C. Plasmids were extracted using the commercial kit (Invitrogen, K210005).

For NGS, 50 ng/rxn original or amplified library plasmids were used for PCR with NGS-F (5'- TTGTGGAAAGGACGAAACACCG-3') and NGS-R (5'-TCTACTATTCTTTCCCCTG-CACTGT-3') primers using PrimeSTAR Max DNA polymerase (Takara, R045Q). Each library was set up three amplification reactions. PCR was carried out under the following conditions: 98˚C for 2 min (initial denaturation); 28 cycles: 98˚C for 15s (denaturation), 53˚C for 15s (primer annealing), 72˚C for 6s (extension); 72˚C for 2 min (final extension). Product was purified using 2% agarose gel using commercial kit (Qiagen, 28704), followed by sequencing using NovaSeqPE250 platform (Novogene Co., Ltd.).

## CRISPR screening

CRISPR screens were performed according to the protocol as previously reported [60]. HeLa cells were stably transduced with lentivirus from LentiCas9-Blast and 10μg/mL polybrene (Millipore, TR-1003-G), and subsequently selected using 10μg/mL Blasticidin (Invivogen, ant-bl). A total of $4x10^7$ HeLa-Cas9 cells were transduced with the lentiviruses of human genome-wide sgRNA library Brubello (Addgene, 73178) at a MOI of 0.3 TU/cell and following by selection using 1μg/mL puromycin (Invivogen, ant-pr) for 7 days. A total of $4x10^7$ mutant cells were transduced with sc-AAV2-CB-EGFP or sc-AAV2-EF1-EGFP at a MOI of 1000vgs/cell. Cells were collected and fixed with 4% PFA (Sigma, 158127). EGFP+ and EGFP- cells were sorted using a BD FACS AriaIII cytometer. Total $4x10^7$ sorted cells of each pupation were treated with 1M Tris-HCl (pH7.5) at 55˚C overnight following by genomic DNA extraction using the commercial kit (Tiangen, DP304).

For NGS, all the genomic DNA were amplified by PCR with NGS-F and NGS-R primers using Phanta Max Super-Fidelity DNA polymerase (Vazyme, P505). PCR was carried out under the following conditions: 95˚C for 3min (initial denaturation); 23 cycles: 95˚C for 1min (denaturation), 58˚C for 30s (primer annealing), 72˚C for 1min (extension); 72˚C for 5min (final extension). Product was purified using 2% agarose gel using the commercial kit (Qiagen, 28704), followed by the preparation of NGS library using the commercial kit (Vazyme, ND607). NovaSeqPE250 platform was used to perform NGS (Novogene Co., Ltd.).

After sequencing, raw reads were mapped to known sgRNA sequences using Bowtie2 and then analyzed using the MAGeCK (v0.5.9.4) pipeline [61,62,63]. The EGFP- population was defined as "control", while the EGFP+ population was defined as "treatment". Significance values of sgRNAs and genes were determined after reads normalization to non-targeting sgRNAs in the library.

## Gene silencing by RNAi and CRISPRi

For RNAi, shRNA sequences were cloned from the RNAi consortium shRNA library [64]. Sense and anti-sense DNA oligonucleotides (Sangon Biotech (Shanghai) Co., Ltd) were annealed and ligated into pLKO.1-TRC. Lentiviral vectors containing shRNAs were packaged

and transduced into HeLa cells with 10μg/mL polybrene. After selection with 1μg/mL puromycin for a week, cells were collected for the following experiments.

For gene silencing in HepG2 or BJ cells, fresh lentiviral supernatant was added with 10μg/mL polybrene. Culture media were changed 24hrs post infection and cells were collected for the following experiments 48hrs post infection.

For CRISPRi [65], sgRNAs were designed using the online tool CRISPROR [66] to target within -50 to +200 bp region near the *SMCHD1* transcription start site (TSS). DNA oligonucleotides were annealed and ligated into pPB-CRISPRi. Transposon and hyPBase-expression plasmid were both transfected into HeLa cells at 3:1 ratio. 24hrs post transfection, media were changed with complete culture medium containing 10μg/mL Blasticidin and cultured for a week. Cells were collected for the following experiments. All oligonucleotides used for cloning can be found in **S1 and S2 Tables**.

### Generation of the *SMCHD1*-KO cell lines

DNA oligonucleotides containing gRNA sequence targeting to *SMCHD1* genomic loci were annealed and ligated into pPB-CRPISPR-copGFP or pPB-CRISPR-mCherry. All the six plasmids were transfected into HeLa cells using Lipofectamine 3000 (Invitrogen, L3000008). After 2 days, copGFP+/mCherry+ cells were sorted, and single clones were cultured. After several weeks, genomic DNA was extracted from cell clones using the commercial kit (Tiangen, DP304) according to the manufacturer's introduction. Knockout of *SMCHD1* was validated by Western Blotting. All oligonucleotides used for cloning can be found in **S3 Table**.

### Cell binding assay

The indicated cell lines were seeded on 24-well plates at $1\times10^5$ cells/well overnight. Cells were placed on ice for 30 min, and then $1\times10^8$ vgs/well scAAV2-CB-EGFP was added. Vectors were allowed to bind cells on ice for 1hr. Flowing binding, cells were washed three times with ice-cold PBS and then 200μL/well lysis buffer was added for genomic DNA extraction using the commercial kit. Viral genome was quantified by qPCR. Normalization of the data was performed in comparison with the level of *ACTIN*. All primer sequences can be found in **S4 Table**.

### Cell entry assay

The indicated cell lines were seeded on 24-well plates at $1\times10^5$ cells/well overnight. Cells were placed on ice for 30min, and then $1\times10^8$ vgs/well scAAV2-CB-EGFP was added. Vectors were allowed to bind cells on ice for 1hr. Flowing binding, cells were washed three times with ice-cold PBS and then cultured at 37°C for 1hr. 1M NaCl was used to strip any residual vectors binding on the cell membrane. Cells were collected using 0.25% trypsin and lysed using 200μL/well lysis buffer for genomic DNA extraction using the commercial kit. Viral genome was quantified as described as above.

### Nucleic entry assay

The indicated cell lines were seeded on 24-well plates at $1\times10^5$ cells/well, $1\times10^8$ vgs/well scAAV2-CB-EGFP was added, and then were cultured for 48hrs. Cells were then collected using 0.25% trypsin and nucleus were isolated using nuclear and cytoplasmic protein extraction kit (Beyotime, P0028). Genomic DNA was extracted using the commercial kit. Viral genome was quantified as described as above.

## Quantification of intracellular AAV genome copies

The indicated cell lines were seeded on 24-well plates at $1 \times 10^5$ cells/well, $1 \times 10^8$ vgs/well scAAV2-CB-EGFP was added, and then were cultured for 48hrs. Cells were then washed three times with ice-cold PBS and then 200μL/well lysis buffer was added for genomic DNA extraction using the commercial kit. Viral genome was quantified as described as above.

## RT-qPCR

For quantification of gene expression, RNA was extracted using TRIzol (Sigma, T9424) according to the manufacturer's introduction. mRNA was reverse-transcribed using the commercial kit (Takara, RR037). After reverse transcription, quantitative PCR was performed using Bio-Rad CFX384 system and gene expression was normalized to cellular *GAPDH* level. All used primer sequences could be found in **S4 Table**.

## Flow cytometry

The indicated cell lines were seeded on 24-well plates at $1 \times 10^5$ cells/well and transduced with indicated rAAVs. At 48 h post transduction, cells were trypsinized and analyzed by flowcytometry using a BD LSR Fortessa cytometer. At least 10,000 events were recorded per sample, and cells were gated based on forward scatter/side scatter (FSC/SSC), FSC-height/FSC-area (FSA-H/FSC-A, singlets), and finally phycoerythrin/fluorescein isothiocyanate (PE/FITC) or fluorescein isothiocyanate/phycoerythrin (FITC/PE) to determine the percentage of infected cells using FlowJo 10 software. In all experiments, uninfected cells were served as a negative control for gate setting.

## Western blotting

Cells were washed twice with PBS and scraped from the surface in PBS. After centrifugation, total proteins were extracted from the cell pellet using RIPA buffer (Beyotime, P0013B) supplemented with protease inhibitor cocktail (Sigma, P8340) by ultrasonication. Protein concentration was quantified using BCA assay (Pierce, 23227). Protein samples were denaturing using SDS-PAGE loading buffer (Solarbio, P1040) and stored at -20˚C.

Samples containing 10–20μg of total protein were loaded on SDS-PAGE and were transferred to 0.45 μm PVDF membrane (Millipore, IPVH00010) after electrophoresis using Bio-Rad SD Semi-Dry transfer cell. Then, membranes were incubated at room temperature for 1hr in blocking buffer (5% non-fat milk in TBS/T). Next, membranes were transferred to primary antibody solution (diluted using 5% BSA in TBS/T) for incubating overnight at 4˚C. The following day, membranes were washed using TBS/T (three times, 5min) and then incubated with secondary antibody diluted in blocking buffer at room temperature for 1hr. After washing in TBS/T (three times, 5min), chemiluminescent substrate (Affinity, KF8005) was added to membranes and the signal was recorded using a Bio-Rad ChemiDoc Imaging System device. For the detection of loading control, membranes were washed using TBS/T and incubated with HRP-conjugated anti-beta Actin antibody (HuaBio, ET1702-67) diluted in blocking buffer at room temperature for 1hr and washed in TBS/T (three times, 5min). After adding substrate, the signal was recorded.

Information of antibodies and dilution ratios used for experiments were listed here: anti-FLAG (Sigma, F1804), 1:2500; anti-SMCHD1 (Abways, CY8117), 1:2000; anti-SP1 (Santa Cruz, sc-59), 1:200; anti-GAPDH (Abways, AB0037), 1:5000; anti-LRIF1 (Proteintech, 26115-1-AP), 1:1000; anti-HP1α (Abways, Cy5502; Abcam, ab109028), 1:2000; Anti-acetyl-Lycine (Immunoway, YM3449), 1:1000; anti-H3 (Abways, CY6587), 1:2000; anti-HP1 (Santa Cruz,

sc-515341), 1:200; HRP-conjugated donkey-anti-mouse IgG (Boster, BA1062), 1:10000; HRP-conjugated goat-anti-rabbit IgG (Boster, BA1054), 1:10000; HPR-conjugated mouse-anti-rabbit IgG HCS (IPkine, A25122), 1:1000; HRP-conjugated rabbit-anti-mouse IgG LCS (IPkine, A25012), 1:1000.

## Immunoprecipitation

For co-immunoprecipitation of AAV genome with FLAG-SMCHD1, $7.5 \times 10^5$ HEK293 cells/well were seeded in 6-well plates overnight. Next, 2.5μg/well of pPB-EF1-FLAG-SMCHD1 or pPB-EF1-BSD plasmids were transfected into cells and following by infecting $2 \times 10^9$ vgs/well of scAAV2-CB-EGFP 24hrs post transfection. Cells were collected after washing with PBS three times and stocked at -80°C.

For co-immunoprecipitation of AAV genome with H3K9me3 histones, H3K27me3 histones, and HP1α, $5 \times 10^6$ HeLa or its derivative cells were seeded in P100 plates and infected with $5*10^9$ vgs/plate of scAAV2-CB-EGFP. Cells were collected after washing with PBS three times and stocked at -80°C.

For co-immunoprecipitation of FLAG-LRIF1, m1, m2, and HA-HP1α, $7.5 \times 10^5$ HEK293 cells/well were seeded in 6-well plates overnight. Next, 2.5μg/well of plasmids were transfected into cells. Cells were collected after washing with PBS three times at 48hrs post transfection and stocked at -80°C.

The immunoprecipitation procedure was as follows: total proteins were resuspended using lysis buffer (NP-40 lysis buffer) (Beyotime, P0013F) supplemented with protease inhibitor cocktail) by incubating on ice for 30min (inverting tubes gently every 5min). After centrifugation at 4°C for 30min at 12000g, the supernatant was collected for quantification using the BCA method. 1mg/sample total protein was adjusted to the volume of 500 μL using lysis buffer. 15μL/sample protein-A/G magnetic beads (MCE, HY-K0202) was added to tubes after washing with NP-40 lysis buffer three times and rotated at 4°C for 30min. Beads were removed using magnetic stand. Next, primary antibodies or equal amount of IgG were added and rotated at 4°C overnight. 25μL/sample pre-cleaned protein-A/G magnetic was added and rotated at 4°C for 4hrs the next following day. Antibody and protein-bound beads were then collected using a magnetic stand and washed with NP-40 lysis buffer three times. For DNA detection, 20μL/sample NP-40 lysis buffer was used to resuspend beads. 2μL/sample beads were diluted using 18μL/sample DNA preparation buffer (0.45mg/mL Protease K in TE buffer) and incubated at 55°C for 1hr, subsequently at 95°C for 10min. After removing beads, samples were used for PCR (EGFP-F: 5'- GAGCGCACCATCTTCTTCAA-3', bGH-pA-R: 5'- CAAC-TAGAAGGCACAGTCGAGG-3') or qPCR (primer sequences were listed in S4 Table). For protein detection, the remaining beads were mixed with 20μL/sample 2x SDS-PAGE loading buffer and denaturing at 95°C for 10min. After removing beads, samples were used for Western Blotting.

Information of antibodies and dilution ratios used for experiments were listed here: anti-FLAG (Sigma, F1804), 2μg/sample; anti-H3K9me3 (CST, 13969), 1:50; anti-H3K27me3 (CST, 9733), 1:150; anti-HP1α (Abways, Cy5502), 1:100; anti-HP1α (Abcam, ab109028), 1μg/sample; anti-HA (CST, 3724), 1:100; mouse normal IgG (Santa Cruz, sc-2025); rabbit IgG, (Abcam, ab172730).

## Bisulfate sequencing

HeLa cells were seeded on 24-well plates at $1 \times 10^5$ cells/well and transduced with $2 \times 10^9$ vgs/well scAAV2-CB-EGFP. Cells were collected for extraction of genomic DNA 48hrs post infection. 15μg of genomic DNA was used to perform bisulfate conversion using DNA methylation kit

(Zymo Research, D5005) according to the manufacturer's introduction. Then, the DNA was amplified by PCR and products were cloned into pUC19. Following transformation, single colonies were selected for sanger sequencing using M13-F primer. All primer sequences can be found in **S5 Table**.

## Luciferase assays

Luciferase assays were performed using commercial kit (Beyotime, RG005) 2 days post transduction according to the manufacturer's introduction. Briefly, cell lysis was prepared using 200μL/well lysis buffer after washing with PBS. After centrifugation, 40μL/well supernatant was added to the 96-well luminescence detection plate (WHB, WHB-96-01). Substrate was added and luminescence was measured using an BioTek Synergy H1 microplate reader.

## DNAscope

Cells were seeded in chamber slides (Millipore, PEZGS0816) and transduced with ssAAV2-C-B-Luc (MOI = 1000 vgs/cell). Cells were then fixed using neutral formalin (Solarbio, G2161) for 30min at 48hrs post transduction. DNAscope was performed for detection of the co-locolization between SMCHD1/vDNA or HP1/vDNA in cells using commercial kit (ACDBio, 323100) according to the manufacturer's introduction. Probe targeting the anti-sense strand of luciferase gene (ACDBio, 852291) was chosen to evade interference from mRNA.

## Immunofluorescence staining

Cells were seeded on glass coverslips cultured for 24hrs in 24-well plates. Cells were then fixed with 4% PFA for 10min at room temperature, washed twice with PBS, next permeabilized using 0.3% Triton X-100 (Sigma, T9284) in PBS for 10 min at room temperature, and then incubated in blocking buffer (1% BSA, 22.52 mg/ml glycine, and 0.1% Tween-20 in PBS) for 1hr at room temperature. Blocked coverslips were then incubated overnight with primary antibodies dilution using blocking buffer at 4˚C. The next day, coverslips were washed in PBS with 0.1% Tween 20 three times and then incubated with secondary antibodies for 2hrs at room temperature. After washing three times in PBS with 0.1% Tween-20, coverslips were mounted with fluorescence mounting medium (ZSBio, ZLI9557).

Information of antibodies and dilution ratios used for experiments were listed here: anti-FLAG (Sigma, F1804), 1:1000; anti-SMCHD1, (Abcam, Ab122555), 1:200; anti-LRIF1 (Millipore, ABE1008), 1:1000; Anti-HP1 (Santa Cruz, sc-515341), 1:200; anti-H3K9me3 (CST, 13969), 1:2000; anti-H3K27me3 (CST, 9733), 1:2000; anti-rabbit Alexa-488 (Invitrogen, A-11008), 1:1000; anti-mouse Alexa-568 (Invitrogen, A-11004), 1:1000.

## Microscopy and imaging analysis

Images were captured with 100X magnification using an Olympus SpinSR-COMB and processed and analyzed using ImageJ software. For co-localization quantification, nuclear were selected according to DAPI signals and then Pearson's correlation coefficients (PCC) of fluorescence signals for both indicated channels were determined using coloc2 plugin in ImageJ. For quantifying HP1 intensities, both vDNA and HP1 signals in nuclear were extracted and then the mean gray values of HP1 channel on vDNA area were calculated.

## Expression profile analysis of candidate restriction factors

Data of RNAseq were downloaded from HPA (human protein atlas) [67]. Genes were defined as non-expression genes if TPM < 3 or active genes if TPM > 3 [68].

## ATAC-qPCR

For ATAC-qPCR, indicated cells were seeded on 12-well plates at $2\times10^5$ cells/well, $2\times10^8$vgs/well scAAV2-CB-EGFP was added, and then were cultured for 48hrs. ATAC-qPCR were performed according to the protocol as previously reported [69]. Cells were treated with 50μg/mL DNase I (Roche, 14104159001) at 37˚C for 30 min. Next, cells were collected for nuclear extraction. For genome tagmentation, in-house prepared Tn5 was used. DNA was then purified and barcoded by PCR under the following conditions: 72˚C for 5min (gap filling); 98˚C for 30s (initial denaturation); 7 cycles: 98˚C for 10s (denaturation), 65˚C for 30s (primer annealing), 65˚C for 45s (extension); 65˚C for 3min (final extension). After purifying, DNA was used for qPCR detection. The accessibility level of different regions was normalized to accessibility level of *GAPDH*. All primer sequences can be found in **S6 Table**.

## Supporting information

**S1 Fig. Amplification and characterization of the sgRNA library. (A)** Schematic of quality evaluation for the amplified CRIPSR library. Brunello library was amplified according to the Addgene protocol (cat. 73178). Amplicon comprising U6 region, sgRNAs, terminator (TTTTT) was prepared for NGS. Total sequencing reads were defined as (1) Non-expression reads, which there were mismatches in U6 or terminator region; (2) Non-perfect reads, which there were mismatches in sgRNA region; (3) Perfect reads, which were perfectly aligned with expected sequences. Counts of perfect reads were used to evaluate library integrity, abundance of sgRNAs, and skew ratio of sgRNA abundance. **(B)** Quality control of NGS data. The majority of sequencing reads were perfectly aligned in both NGS data. **(C)** Amplified library maintained most of the original sgRNAs. **(D, E)** Distribution of sgRNAs was uniform in the amplified library. The red line in (**D**) indicated the median counts of sgRNAs. Skew ratio was calculated by dividing the 90th percentile by the 10th percentile normalized sgRNA counts. (TIF)

**S2 Fig. Identification of host essential and restriction factors for AAV transduction. (A, B)** EGFP+ and EGFP- populations were sorted from scAAV2 transduced cells. **(C)** Entire library was delivered into cells. Zero-count sgRNAs in EGFP+ and EGFP- cells were less than 1%. **(D, E)** Candidate essential factors of AAV transduction. Candidates were defined as the hits only if outstanding in both scAAV2-CB-EGFP and scAAV2-EF1-EGFP screenings. Data were analyzed by MAGeCK, and the EGFP- group was defined as the control. Hits were defined according to the following indicators calculated by MAGeCK: (1) p < 0.01, (2) LFC < -0.5, and (3) number of good sgRNA > 1. **(F, G)** Information of candidates in both scAAV2-CB-EGFP and scAAV2-EF1-EGFP screenings. LFC, logarithm of fold change with base 2. (TIF)

**S3 Fig. Validation of candidate restriction factors. (A)** RT-qPCR confirmed knockdown of candidate in HeLa. Error bar represented data from three independent experiments. Statistics: One-way ANOVA by SPSS v29.0. \*\*\*p < 0.001, \*\*p < 0.01, \*p<0.05. **(B)** Percentage of EGFP + cells were quantified by FACS. All the eight candidates were separated into three groups for validation. *AP1G1*, *AP1M1*, and *HEATR5B* for group 1. *SMCHD1*, *UBE2N*, and *YPEL3* for group 2. *NELFCD* and *FAM122A* for group 3. Representative data from FACS were shown. (TIF)

**S4 Fig. *SMCHD1*-KD enhanced AAV2 transduction in both cell lines and primary fibroblasts. (A)** RT-qPCR confirmed knockdown of *SMCHD1* in HepG2. Error bar represented data from two independent experiments. **(B)** *SMCHD1*-KD significantly enhanced

scAAV2-CB-EGFP transduction in HepG2 cells. **(C)** RT-qPCR confirmed knockdown of *SMCHD1* in BJ. Error bar represented data from two independent experiments. **(D)** *SMCHD1*-KD significantly enhanced scAAV2-CB-EGFP transduction in BJ cells.
(TIF)

**S5 Fig. Co-localization of SMCHD1/vDNAs was detected in both HeLa and primary fibroblasts. (A)** Probe for luciferase was specific to vDNAs in HeLa cells. Both transduced and untransduced HeLa cells were fixed and immunostaining with anti-SMCHD1 antibody and luciferase DNA-targeting probe. Scale bars: 10μm. **(B)** Probe for luciferase was specific to vDNAs in BJ cells. Scale bars: 10μm. **(C)** Co-localization of endogenous SMCHD1 with AAV DNAs in BJ cells. Scale bars: 5μm and 1μm (zoom-in).
(TIF)

**S6 Fig. *SMCHD1*-KO disrupted the co-localization between LRIF1 and HP1.** Co-localization of LRIF1 and HP1 were detected by IF. Yellow arrowheads, co-localization of LRIF1 and HP1. White arrowheads, disrupted co-localization of LRIF1 and HP1. Scale bar: 5μm. Co-localization was quantified using PCC methods by coloc2 plugin in Image J. Each dot represents a single cell. Statistics: One-way ANOVA by SPSS v29.0. ***$p < 0.001$.
(TIF)

**S7 Fig. SMCHD1 knockout did not affect methylation status of the AAV genome and its associated histones. (A)** Lack of intense DNA methylation in AAV genomic regions. The transcriptional regulating elements in the scAAV2-CB-EGFP genome included a CMV enhancer, a chicken beta-globin promoter, and a SV40 intron from 5' to 3'. HeLa cells were transduced with scAAV2-CB-EGFP (MOI: 20000 vgs/cell), and the genomic DNA was extracted for bisulfate conversion. Amplicons covering the regulating elements were cloned for sanger sequence. Each circle indicates a CpG dinucleotide. Solid circle means the methylated CpG site while hollow circle indicates the unmethylated CpG dinucleotide. *SOX11* promoter sequence was used as a positive control for CpG methylation. **(B)** *SMCHD1*-KO did not alter the overall H3K9me3 and H3K37me3 modifications in nucleus. (Scale bar: 10μm). **(C)** Immunoprecipitation of H3K9me3 and H3K27me3. Hela cells were lysed using RIPA buffer and protein samples were used for immunoprecipitating. Normal IgG worked as control. (D) *SMCHD1*-KO did not affect H3K9me3 and H3K37me3 modifications on AAV genome. HeLa or *SMCHD1*-KO cell lysate (NP-40 lysis buffer) transduced with scAAV2-CB-EGFP (MOI: 1000 vgs/cell) were used to test the levels of H3K9me3 and H3K37me3 modifications on AAV genome by immunoprecipitatition with anti-H3K9me3 and anti-H3K37me3 antibodies. Normal IgG was used as control. Level of co-immunoprecipitated AAV genome DNA was quantified by qPCR. Error bar represented data from three independent experiments. Statistics: One-way ANOVA by SPSS v29.0, n.s., not significant.
(TIF)

**S1 Table. Sequence of oligonucleotides used for shRNA cloning in pLKO.1-TRC.**
(XLSX)

**S2 Table. Sequence of oligonucleotides used for CRISPRi cloning.**
(XLSX)

**S3 Table. Sequence of oligonucleotides used for *SMCHD1*-KO.**
(XLSX)

**S4 Table. Primers used for qPCR.**
(XLSX)

**S5 Table. Primers used for bisulfate sequencing.**
(XLSX)

**S6 Table. Primers used for ATAC-qPCR.**
(XLSX)

## Acknowledgments

We would like to thank the Core Facilities in the College of Life Sciences for their technical assistance.

## Author Contributions

**Conceptualization:** Biao Dong, Zhonghan Li.

**Data curation:** Chenlu Wang, Yu Liu, Kun Xie, Yu Hu, Baiquan Zhang, Xiaochao Huang, Hui Bao.

**Formal analysis:** Chenlu Wang, Yu Liu, Jingfei Xiong, Tianshu Wang, Yu Hu, Huancheng Fu.

**Funding acquisition:** Haoyang Cai, Zhonghan Li.

**Investigation:** Chenlu Wang.

**Methodology:** Chenlu Wang, Yu Liu, Jingfei Xiong, Tianshu Wang, Baiquan Zhang, Xiaochao Huang, Hui Bao.

**Project administration:** Zhonghan Li.

**Resources:** Zhonghan Li.

**Supervision:** Haoyang Cai, Biao Dong, Zhonghan Li.

**Validation:** Chenlu Wang, Kun Xie, Huancheng Fu, Hui Bao.

**Visualization:** Chenlu Wang, Jingfei Xiong, Tianshu Wang, Huancheng Fu.

**Writing – original draft:** Chenlu Wang, Zhonghan Li.

**Writing – review & editing:** Haoyang Cai, Biao Dong, Zhonghan Li.

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
