## [Decision Letter · Decision Letter 0]

16 Feb 2024

Dear Dr. Li

Thank you very much for submitting your manuscript "Genome-wide CRISPR screenings identified SMCHD1 as a host-restricting factor for AAV transduction" for consideration at PLOS Pathogens. As with all papers reviewed by the journal, your manuscript was reviewed by members of the editorial board and by several independent reviewers. In light of the reviews (below this email), we would like to invite the resubmission of a significantly-revised version that takes into account the reviewers' comments.  All three reviewers raised similar concerns regarding the mechanistic aspects of your manuscript, and pointed out that this manuscript would be much stronger if you directly tested your hypothesis by performing key experiments outlined in major experiments sections by all three reviewers.  Further, it is also important that you validate the identified direct interactions (with CO-IP or other methods).  Finally, the addition of proper controls for experiments outlined in figures 3-5 are also warranted.  Please note that it is important for you to address all of the reviewers concerns in both the response to reviewers document as well as the manuscript.

We cannot make any decision about publication until we have seen the revised manuscript and your response to the reviewers' comments. Your revised manuscript is also likely to be sent to reviewers for further evaluation.

Sincerely,

Donna M Neumann

Academic Editor

PLOS Pathogens

Blossom Damania

Section Editor

PLOS Pathogens

Michael Malim

Editor-in-Chief

PLOS Pathogens

orcid.org/0000-0002-7699-2064

Reviewer's Responses to Questions

**Part I - Summary**

Reviewer #1: In this manuscript, Wang et al. use genome-wide CRISPR screening to identify host factors that restrict AAV transgene expression. As low transgene expression is thought to be a significant contributor to the high viral loads needed for AAV in gene therapy applications, an improved understanding of the mechanisms regulating AAV transgene expression is important for devising potential strategies to improve therapeutic approaches.

In general, the manuscript is clearly written and the data, for the most part, are well controlled, robust, and compelling. The identification and validation of SMCHD1 as a restriction factor for AAV gene expression is especially thorough, using multiple ways of targeting SMCHD1 and multiple cell types and AAV serotypes. The authors go on to characterize the mechanism by which SMCHD1 represses AAV transgene expression, concluding it does so through a complex with LRIF1 and HP1, likely resulting in condensation and repression of viral chromatin. Unfortunately, the authors do not directly test key aspects of their mechanistic model, namely that SMCHD1 and LRIF1 repress AAV gene expression via HP1 and chromatin condensation, leaving the mechanistic aspect of the manuscript somewhat incomplete.

Reviewer #2: In this manuscript, Wang et al perform a genome wide CRISPR screen to determine host factors that limit AAV transduction efficiency. Among other factors, they discover that SMCHD1 restricts AAV gene expression after nuclear entry in multiple cell types and with multiple serotypes of AAV. Overall, these findings are rigorously explored using multiple techniques. However, in multiple instances certain conclusions are over-interpreted or at least could be supported with alternative methodologies or additional quantification as detailed below.

Reviewer #3: In this manuscript, the authors performed a genome-wide CRISPR screen to identify factors that restrict adeno-associated virus (AAV) transduction. They identified the epigenetic modifier SMCHD1 (among others) as crucial protein that impairs AAV gene transduction. They could demonstrate that SMCHD1 does not affect AAV attachment, entry or delivery of the genome to the nucleus, but restricts transgene expression. In addition, SMCHD1 directly binds the AAV genome and forms as complex with LRIF1 and HP1, leading to the heterochromatinization of the AAV genome. Overall, this is a well written manuscript and could contribute to the improvement in AAV gene delivery. However, a few points should be addressed prior to publication.

**Part II – Major Issues: Key Experiments Required for Acceptance**

Reviewer #1: 1. The authors should test HP1 association with viral DNA in SMCHD1 and/or LRIF1 deficient cells. Their model suggests that SMCHD1/LRIF1 suppress AAV2 gene expression via HP1 and chromatin condensation, but they do not present experiments that directly test this. The data shown in Fig. 5 show that HP1 is associated with viral genomes in wild-type cells, but do not test whether that association is abrogated or diminished in the absence of SMCHD1 and/or LRIF1.

Similarly, are the HP1 co-localization experiments shown in Fig. 5 and Fig. S8 performed in transduced or untransduced cells? Can the authors show that HP1 co-localizes with viral DNA and, if so, whether loss of SMCHD1 and/or LRIF1 disrupts that localization?

2. Is HP1 binding required for repression by SMCHD1 / LRIF1? Can the authors show whether reconstitution of LRIF1 KD cells with a HP1 or SMCHD1 binding mutant (ref. 43) fails to rescue repression of AAV gene expression? Such a result would strengthen the authors’ mechanistic conclusions.

Reviewer #2: 1. In Figure 3 a direct link is claimed between SMCHD1 and AAV genomes. However the standard PCR performed in Figure 3C is not quantitative or quantitated and is lacking information about the percentages of material in either the input or the IP. Furthermore, one field of microscopy data is shown highlighting a single overlap of AAV genome and SMCHD1. This key experiment needs to be quantified over multiple fields and multiple replicates. Ideally the same experiment would also be performed on uninfected cells to highlight any potential changes in SMCHD1 morphology as well as the cleanliness of highly-amplified DNAscope experiments.

2. In Figure 4A need quantification over multiple fields and multiple replicates for interaction between SMCHD1 and LRIF1. Alternatively, biochemical interaction could be shown with co-immunoprecipitation.

3. In Figure 5A again a physical interaction is implied between both SMCHD1, LRIF1, and HP1 via one field of view of microscopy. Especially in Figure 5D where loss of LRIF1 is claimed to stop interaction of SMCHD1 with HP1 this should be validated with co-immunoprecipitation.

4. The authors final model claims that SMCHD1-HP1 is condensing AAV genomes into an inactive state, however no experimentation has been done to support this hypothesis. It is fine to speculate about this but should be saved for the discussion.

Reviewer #3: 1) The authors show the colocalizations between the AAV genome, SMCHD1 and other proteins, but it remains elusive in what percentage of cells this occurs. Does the AAV genome and SMCHD1 colocalize in 100 % of these cells (Fig 3d)? If yes, state how many cells were analyzed. If not then properly quantify the colocalization. The same is true for the colocalization of SMCHD1, LRIF1 and HP1. Does the AAV genome and SMCHD1 also colocalize in other cell types?

**Part III – Minor Issues: Editorial and Data Presentation Modifications**

Reviewer #1: 1. Does repopulation of SMCHD1 KO cells with SMCHD1 reverse the enhancement of AAV2 gene expression? Similar to what the authors have shown for repopulation with LRIF1 in Fig. 4g? The authors have robustly shown SMCHD1 deficiency results in increased AAV gene expression, but demonstrating they can reverse that by re-introduction of SMCHD1 would further bolster this conclusion.

2. Fig 3e: the authors should include an untransduced control to demonstrate the specificity of their AAV DNAscope. It would also be helpful to quantitate this and other microscopy data such as Fig. 4a, 5a, and 5d to show the degree to which colocalization does or does not occur across the population of cells.

3. Is repression by SMCHD1 dependent on LRIF1? The authors demonstrate in Fig. 4 that repression by LRIF1 is dependent on SMCHD1, but because the data in Fig 4f appears to be normalized to the Scr condition for each cell line, it is impossible to evaluate whether enhancement of AAV2 gene expression by SMCHD1 knockout is dependent on LRIF1.

4. Fig 2. Panels a-e (SMCHD1 KO and KD in additional cell types) seem like they could better fit in Fig. 1 establishing SMCHD1 as an inhibitor of AAV2 transgene expression vs. in Fig 2 which is testing multiple serotypes of AAV.

5. Figure legend for panel 3a references a western blot which is not present. I think this may actually be Fig. S7?

6. Figure legend for Fig. 4: panels e and f appear to be swapped.

Reviewer #2: 1. Panel labeling in Figure 4 is off compared to the text.

2. Line 271 is confusing (experiment using FLAG-tagged LRIF1 was carried out and supplementing the shRNA-resistant LRIF1 would abolish the AAV transcription enhancement in LRIF1-KD cells). Regardless, the data presented in Figure 4G shows a statistical decrease in the RLU after adding back in shRNA-resistant LRIF1 as described, but statistics are not performed to determine shLRIF1 increased RLU over scrambled.

3. Figure 5C y-axis labeling is confusing. Why not simply use % Input as is done in S9.

4. Overall the abundance of supplemental figures made this manuscript harder to read. In a journal with no prescribed figure limits it would make sense to bring at least some of this data into a more easily accessible area.

Reviewer #3: 1) Figure 4: the labels of the figure are partially mixed up. E.g. “...the enhancement was largely abolished in SMCHD1-KO cells (Fig. 4e)” but this is shown in panel 4f. The same is true for “...experiment using FLAG-tagged LRIF1... (Fig. 4f, g)”. The Flag tag data is currently shown in Fig. 4e and 4g. Please correct and double check all figures.

2) The color scheme of Figure 4f and 4g is confusing. The LRIF1 shRNA #2 is shown in purple in 4f. So why is also the scrambled shRNA purple in 4g too? The controls were shown in grey in the other figures.

3) In Fig. 4g the legend for the symbols (circle or diamond) is missing.

4) In the summary figure 5g, it is not obvious at the first glance that the left half is the wild type scenario and the right is the SMCHD1 knockout one. Please address this by e.g. increase the fond size of these labels to make it obvious to all readers.

PLOS authors have the option to publish the peer review history of their article (what does this mean?). If published, this will include your full peer review and any attached files.

Reviewer #1: No

Reviewer #2: No

Reviewer #3: No
---

## [Editor Report · Decision Letter 1]

14 Jun 2024

Dear Dr. Li

We are pleased to inform you that your manuscript 'Genome-wide CRISPR screenings identified SMCHD1 as a host-restricting factor for AAV transduction' has been provisionally accepted for publication in PLOS Pathogens.

Best regards,

Donna M Neumann

Academic Editor

PLOS Pathogens

Blossom Damania

Section Editor

PLOS Pathogens

Michael Malim

Editor-in-Chief

PLOS Pathogens

orcid.org/0000-0002-7699-2064
---

## [Editor Report · Acceptance letter]

28 Jun 2024

Dear Prof. Li,

We are delighted to inform you that your manuscript, "Genome-wide CRISPR screenings identified SMCHD1 as a host-restricting factor for AAV transduction," has been formally accepted for publication in PLOS Pathogens.

Best regards,

Michael Malim

Editor-in-Chief

PLOS Pathogens

orcid.org/0000-0002-7699-2064